# Pessimistic Bootstrapping for Uncertainty-Driven Offline Reinforcement Learning

**Chenjia Bai**
Harbin Institute of Technology
`baichenjia255@gmail.com`

**Lingxiao Wang**
Northwestern University

**Zhuoran Yang**
Princeton University

**Zhihong Deng**
University of Technology Sydney

**Animesh Garg**
University of Toronto
Vector Institute, NVIDIA

**Peng Liu**
Harbin Institute of Technology

**Zhaoran Wang**
Northwestern University

## Abstract

Offline Reinforcement Learning (RL) aims to learn policies from previously collected datasets without exploring the environment. Directly applying off-policy algorithms to offline RL usually fails due to the extrapolation error caused by the out-of-distribution (OOD) actions. Previous methods tackle such problems by penalizing the Q-values of OOD actions or constraining the trained policy to be close to the behavior policy. Nevertheless, such methods typically prevent the generalization of value functions beyond the offline data and also lack a precise characterization of OOD data. In this paper, we propose Pessimistic Bootstrapping for offline RL (PBRL), a purely uncertainty-driven offline algorithm without explicit policy constraints. Specifically, PBRL conducts uncertainty quantification via the disagreement of bootstrapped Q-functions, and performs pessimistic updates by penalizing the value function based on the estimated uncertainty. To tackle the extrapolating error, we further propose a novel OOD sampling method. We show that such OOD sampling and pessimistic bootstrapping yields a provable uncertainty quantifier in linear MDPs, thus providing the theoretical underpinning for PBRL. Extensive experiments on D4RL benchmark show that PBRL has better performance compared to the state-of-the-art algorithms.

## 1 Introduction

Deep Reinforcement Learning (DRL) (Sutton & Barto, 2018) achieves remarkable success in a variety of tasks. However, in most successful applications, DRL requires millions of interactions with the environment. In real-world applications such as navigation (Mirowski et al., 2018) and healthcare (Yu et al., 2019), acquiring a large number of samples by following a possibly suboptimal policy can be costly and dangerous. Alternatively, practitioners seek to develop RL algorithms that learn a policy based solely on an offline dataset, where the dataset is typically available. However, directly adopting online DRL algorithms to the offline setting is problematic. On the one hand, policy evaluation becomes challenging since no interaction is allowed, which limits the usage of on-policy algorithms. On the other hand, although it is possible to slightly modify the off-policy value-based algorithms and sample solely from the offline dataset in training, such modification typically suffers from a significant performance drop compared with their online learning counterpart (Levine et al., 2020). An important reason for such performance drop is the so-called *distributional shift*. Specifically, the offline dataset follows the visitation distribution of the *behavior policies*. Thus, estimating the $Q$-functions of the corresponding greedy policy with the offline dataset is biased due to the difference in visitation distribution. Such bias typically leads to a significant *extrapolation error* for DRL algorithms since the estimated $Q$-function tends to overestimate the out-of-distribution (OOD) actions (Fujimoto et al., 2019).

To tackle the distributional shift issue in offline RL, previous successful approaches typically fall into two categories, namely, policy constraints (Kumar et al., 2019; Fujimoto & Gu, 2021) and conservative methods (Kumar et al., 2020; Yu et al., 2020; 2021). Policy constraints aim to restrict the learned policy to be close to the behavior policy, thus reducing the extrapolation error in policy evaluation. Conservative methods seek to penalize the $Q$-functions for OOD actions in policy evaluation and hinge on a gap-expanding property to regularize the OOD behavior. Nevertheless, since policy constraints explicitly confine the policy to be close to the behavior policy, such method tends to be easily affected by the non-optimal behavior policy. Meanwhile, although the conservative algorithms such as CQL (Kumar et al., 2020) do not require policy constraints, CQL equally penalizes the OOD actions and lacks a precise characterization of the OOD data, which can lead to overly conservative value functions. To obtain a more refined characterization of the OOD data, uncertainty quantification is shown to be effective when associated with the model-based approach (Yu et al., 2020; Kidambi et al., 2020), where the dynamics model can be learned in static data thus providing more stable uncertainty in policy evaluation. Nevertheless, model-based methods need additional modules and may fail when the environment becomes high-dimensional and noisy. In addition, the uncertainty quantification for model-free RL is more challenging since the $Q$-function and uncertainty quantifier need to be learned simultaneously (Yu et al., 2021).

To this end, we propose Pessimistic Bootstrapping for offline RL (PBRL), an uncertainty-driven model-free algorithm for offline RL. To acquire reliable $Q$-function estimates and their corresponding uncertainty quantification, two components of PBRL play a central role, namely, bootstrapping and OOD sampling. Specifically, we adopt bootstrapped $Q$-functions (Osband et al., 2016) for uncertainty quantification. We then perform pessimistic $Q$-updates by using such quantification as a penalization. Nevertheless, solely adopting such penalization based on uncertainty quantification is neither surprising nor effective. We observe that training the $Q$-functions based solely on the offline dataset does not regularize the OOD behavior of the $Q$-functions and suffers from the extrapolation error. To this end, we propose a novel OOD sampling technique as a regularizer of the learned $Q$-functions. Specifically, we introduce additional OOD datapoints into the training buffer. The OOD datapoint consists of states sampled from the training buffer, the corresponding OOD actions sampled from the current policy, and the corresponding OOD target based on the estimated $Q$-function and uncertainty quantification. We highlight that having such OOD samples in the training buffer plays an important role in both the $Q$-function estimation and the uncertainty quantification. We remark that, OOD sampling controls the OOD behavior in training, which guarantees the stability of the trained bootstrapped $Q$-functions. We further show that under some regularity conditions, such OOD sampling is provably efficient under the linear MDP assumptions.

We highlight that PBRL exploits the OOD state-action pairs by casting a more refined penalization over OOD data points, allowing PBRL to acquire better empirical performance than the policy-constraint and conservatism baselines. As an example, if an action lies close to the support of offline data but is not contained in the offline dataset, the conservatism (Kumar et al., 2020) and policy constraint (Fujimoto et al., 2019) methods tend to avoid selecting it. In contrast, PBRL tends to assign a small $Q$-penalty for such an action as the underlying epistemic uncertainty is small. Hence, the agent trained with PBRL has a higher chance consider such actions if the corresponding value estimate is high, yielding better performance than the policy constraint and conservatism baselines. Our experiments on the D4RL benchmark (Fu et al., 2020) show that PBRL provides reasonable uncertainty quantification and yields better performance compared to the state-of-the-art algorithms.

## 2 PRELIMINARIES

We consider an episodic MDP defined by the tuple $(\mathcal{S}, \mathcal{A}, T, r, \mathbb{P})$, where $\mathcal{S}$ is the state space, $\mathcal{A}$ is the action space, $T \in \mathbb{N}$ is the length of episodes, $r$ is the reward function, and $\mathbb{P}$ is the transition distribution. The goal of RL is to find a policy $\pi$ that maximizes the expected cumulative rewards $\mathbb{E}\big[\sum_{i=0}^{T-1} \gamma^t r_i\big]$, where $\gamma \in [0, 1)$ is the discount factor in episodic settings. The corresponding $Q$-function of the optimal policy satisfies the following Bellman operator,

$$\mathcal{T} Q_\theta(s, a) := r(s, a) + \gamma \mathbb{E}_{s' \sim T(\cdot|s,a)}\big[\max_{a'} Q_{\theta^-}(s', a')\big]. \tag{1}$$

where $\theta$ is the parameters of $Q$-network. In DRL, the $Q$-value is updated by minimizing the TD-error, namely $\mathbb{E}_{(s,a,r,s')}[(Q - \mathcal{T}Q)^2]$. Empirically, the target $\mathcal{T}Q$ is typically calculated by a separate target-network parameterized by $\theta^-$ without gradient propagation (Mnih et al., 2015). In online RL,

one typically samples the transitions $(s, a, r, s')$ through iteratively interacting with the environment. The $Q$-network is then trained by sampling from the collected transitions.

In contrast, in offline RL, the agent is not allowed to interact with the environment. The experiences are sampled from an offline dataset $\mathcal{D}_{\text{in}} = \{(s_t^i, a_t^i, r_t^i, s_{t+1}^i)\}_{i \in [m]}$. Naive off-policy methods such as $Q$-learning suffer from the *distributional shift*, which is caused by different visitation distribution of the behavior policy and the learned policy. Specifically, the greedy action $a'$ chosen by the target $Q$-network in $s'$ can be an OOD-action since $(s', a')$ is scarcely covered by the dateset $\mathcal{D}_{\text{in}}$. Thus, the value functions evaluated on such OOD actions typically suffer from significant extrapolation errors. Such errors can be further amplified through propagation and potentially diverges during training. We tackle such a challenge by uncertainty quantification and OOD sampling.

## 3 PESSIMISTIC BOOTSTRAPPING FOR OFFLINE RL

### 3.1 UNCERTAINTY QUANTIFICATION WITH BOOTSTRAPPING

In PBRL, we maintain $K$ bootstrapped $Q$-functions in critic to quantify the epistemic uncertainty. Formally, we denote by $Q^k$ the $k$-th $Q$-function in the ensemble. $Q^k$ is updated by fitting the following target

$$\widehat{\mathcal{T}} Q_\theta^k(s, a) := r(s, a) + \gamma \widehat{\mathbb{E}}_{s' \sim P(\cdot|s,a), a' \sim \pi(\cdot|s)} \left[ Q_{\theta^-}^k(s', a') \right]. \tag{2}$$

Here we denote the empirical Bellman operator by $\widehat{\mathcal{T}}$, which estimates the expectation $\mathbb{E}[Q_{\theta^-}^k(s', a') \,|\, s, a]$ empirically based on the offline dataset. We adopt such an ensemble technique from Bootstrapped DQN (Osband et al., 2016), which is initially proposed for the online exploration task. Intuitively, the ensemble forms an estimation of the posterior distribution of the estimated $Q$-functions, which yields similar value on areas with rich data and diversely on areas with scarce data. Thus, the deviation among the bootstrapped $Q$-functions yields an epistemic uncertainty estimation, which we aim to utilize as a penalization in estimating the $Q$-functions. Specifically, we introduce the following uncertainty quantification $\mathcal{U}(s, a)$ based on the $Q$-functions $\{Q^k\}_{k \in [K]}$,

$$\mathcal{U}(s, a) := \text{Std}(Q^k(s, a)) = \sqrt{\frac{1}{K} \sum_{k=1}^{K} \left( Q^k(s, a) - \bar{Q}(s, a) \right)^2}. \tag{3}$$

Here we denote by $\bar{Q}$ the mean among the ensemble of $Q$-functions. From the Bayesian perspective, such uncertainty quantification yields an estimation of the standard deviation of the posterior of $Q$-functions. To better understand the effectiveness of such uncertainty quantification, we illustrate with a simple prediction task. We use 10 neural networks with identical architecture and different initialization as the ensemble. We then train the ensemble with 60 datapoints in $\mathbb{R}^2$ plane, where the covariate $x$ is generated from the standard Gaussian distribution, and the response $y$ is obtained by feeding $x$ into a randomly generated neural network. We plot the datapoints for training and the uncertainty quantification in Fig. 1(a). As shown in the figure, the uncertainty quantification rises smoothly from the in-distribution datapoints to the OOD datapoints.

In offline RL, we perform regression $(s, a) \rightarrow \widehat{\mathcal{T}} Q^k(s, a)$ in $\mathcal{D}_{\text{in}}$ to train the bootstrapped $Q$-functions, which is similar to regress $x \rightarrow y$ in the illustrative task. The uncertainty quantification allows us to quantify the deviation of a datapoint from the offline dataset, which provides more refined conservatism compared to the previous methods (Kumar et al., 2020; Wu et al., 2019).

### 3.2 PESSIMISTIC LEARNING

We now introduce the pessimistic value iteration based on the bootstrapped uncertainty quantification. The idea is to penalize the $Q$-functions based on the uncertainty quantification. To this end, we propose the following target for state-action pairs sampled from $\mathcal{D}_{\text{in}}$,

$$\widehat{\mathcal{T}}^{\text{in}} Q_\theta^k(s, a) := r(s, a) + \gamma \widehat{\mathbb{E}}_{s' \sim P(\cdot|s,a), a' \sim \pi(\cdot|s)} \left[ Q_{\theta^-}^k(s', a') - \beta_{\text{in}} \, \mathcal{U}_{\theta^-}(s', a') \right], \tag{4}$$

where $\mathcal{U}_{\theta^-}(s', a')$ is the uncertainty estimation at $(s', a')$ based on the target network, and $\beta_{\text{in}}$ is a tuning parameter. In addition, the empirical mean $\widehat{\mathbb{E}}_{s' \sim P(\cdot|s,a), a' \sim \pi(\cdot|s)}$ is obtained by sampling the

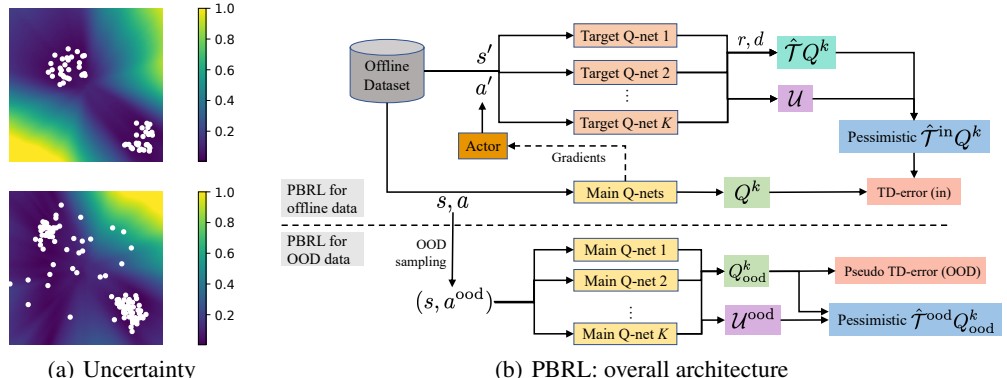

(a) Uncertainty          (b) PBRL: overall architecture

Figure 1: (a) Illustration of the uncertainty estimations in the regression task. The white dots represent data points, and the color scale represents the bootstrapped-uncertainty values in the whole input space. (b) Illustration of the workflow of PBRL. PBRL splits the loss function into two components. The TD-error (in) represents the regular TD-error for in-distribution data (i.e., from the offline dataset), and pseudo TD-error (ood) represent the loss function for OOD actions. In the update of $Q$-functions, both losses are computed and summed up for the gradient update.

transition $(s, a, s')$ from $\mathcal{D}_{\text{in}}$ and further sampling $a' \sim \pi(\cdot \mid s')$ from the current policy $\pi$. We denote by $\widehat{\mathcal{T}}^{\text{in}} Q_\theta^k(s, a)$ the in-distribution target of $Q_\theta^k(s, a)$ and write $\widehat{\mathcal{T}}^{\text{in}}$ to distinguish the in-distribution target from that of the OOD target, which we introduce in the sequel.

We remark that there are two options to penalize the $Q$-functions through the operator $\widehat{\mathcal{T}}^{\text{in}}$. In Eq. (4), we penalize the next-$Q$ value $Q_{\theta-}^k(s', a')$ with the corresponding uncertainty $\mathcal{U}_{\theta-}(s', a')$. Alternatively, we can also penalize the immediate reward by $\hat{r}(s, a) := r(s, a) - \mathcal{U}_\theta(s, a)$ and use $\hat{r}(s, a)$ in place of $r(s, a)$ for the target. Nevertheless, since the datapoint $(s, a) \in \mathcal{D}_{\text{in}}$ lies on rich-data areas, the penalization $\mathcal{U}_\theta(s, a)$ on the immediate reward is usually very small thus having less effect in training. In PBRL, we penalize the uncertainty of $(s', a')$ in the next-$Q$ value.

Nevertheless, our empirical findings in §D suggest that solely penalizing the uncertainty for in-distribution target is insufficient to control the OOD performance of the fitted $Q$-functions. To enforce direct regularization over the OOD actions, we incorporate the OOD datapoints directly in training and sample OOD data of the form $(s^{\text{ood}}, a^{\text{ood}}) \in \mathcal{D}_{\text{ood}}$. Specifically, we sample OOD states from the in-distribution dataset $\mathcal{D}_{\text{in}}$. Correspondingly, we sample OOD actions $a^{\text{ood}}$ by following the current policy $\pi(\cdot \mid s^{\text{ood}})$. We highlight that such OOD sampling requires only the offline dataset $\mathcal{D}_{\text{in}}$ and does not require additional generative models or access to the simulator.

It remains to design the target for OOD samples. Since the transition $P(\cdot \mid s^{\text{ood}}, a^{\text{ood}})$ and reward $r(s^{\text{ood}}, a^{\text{ood}})$ are unknown, the true target of OOD sample is inapplicable. In PBRL, we propose a novel pseudo-target for the OOD datapoints,

$$\widehat{\mathcal{T}}^{\text{ood}} Q_\theta^k(s^{\text{ood}}, a^{\text{ood}}) := Q_\theta^k(s^{\text{ood}}, a^{\text{ood}}) - \beta_{\text{ood}} \, \mathcal{U}_\theta(s^{\text{ood}}, a^{\text{ood}}) \,, \qquad (5)$$

which introduces an additional uncertainty penalization $\mathcal{U}_\theta(s^{\text{ood}}, a^{\text{ood}})$ to enforce pessimistic $Q$-function estimation, and $\beta_{\text{ood}}$ is a tuning parameter. For OOD samples that are close to the in-distribution data, such penalization is small and the OOD target is close to the $Q$-function estimation. In contrast, for OOD samples that are distant away from the in-distribution dataset, a larger penalization is incorporated into the OOD target. In our implementation, we introduce an additional truncation to stabilize the early stage training as $\max\{0, \mathcal{T}^{\text{ood}} Q_\theta^k(s^{\text{ood}}, a^{\text{ood}})\}$. In addition, we remark that $\beta_{\text{ood}}$ is important to the empirical performance. Specifically,

- At the beginning of training, both the $Q$-functions and the corresponding uncertainty quantifications are inaccurate. We use a large $\beta_{\text{ood}}$ to enforce a strong regularization on OOD actions.

- We then gradually decrease $\beta_{\text{ood}}$ in the training process since the value estimation and uncertainty quantification becomes more accurate in training. We remark that a smaller $\beta_{\text{ood}}$ requires more accurate uncertainty estimate for the pessimistic target estimation $\widehat{\mathcal{T}}^{\text{ood}} Q^k$. In addition, a decaying parameter $\beta_{\text{ood}}$ stabilizes the convergence of $Q_\theta^k(s^{\text{ood}}, a^{\text{ood}})$ in the training from the empirical perspective.

Incorporating both the in-distribution target and OOD target, we conclude the loss function for critic in PBRL as follows,

$$\mathcal{L}_{\text{critic}} = \widehat{\mathbb{E}}_{(s,a,r,s') \sim \mathcal{D}_{\text{in}}} \big[ (\widehat{\mathcal{T}}^{\text{in}} Q^k - Q^k)^2 \big] + \widehat{\mathbb{E}}_{s^{\text{ood}} \sim \mathcal{D}_{\text{in}}, a^{\text{ood}} \sim \pi} \big[ (\widehat{\mathcal{T}}^{\text{ood}} Q^k - Q^k)^2 \big], \qquad (6)$$

where we iteratively minimize the regular TD-error for the offline data and the pseudo TD-error for the OOD data. Incorporated with OOD sampling, PBRL obtains a smooth and pessimistic value function by reducing the extrapolation error caused by high-uncertain state-action pairs.

Based on the pessimistic $Q$-functions, we obtain the corresponding policy by solving the following maximization problem,

$$\pi_{\varphi} := \max_{\varphi} \widehat{\mathbb{E}}_{s \sim \mathcal{D}_{\text{in}}, a \sim \pi(\cdot|s)} \Big[ \min_{k=1,\ldots,K} Q^k(s,a) \Big], \qquad (7)$$

where $\varphi$ is the policy parameters. Here we follow the previous actor-critic methods (Haarnoja et al., 2018; Fujimoto et al., 2018) and take the minimum among ensemble $Q$-functions, which stablizes the training of policy network. We illustrate the overall architecture of PBRL in Fig. 1(b).

## 3.3 THEORETICAL CONNECTIONS TO LCB-PENALTY

In this section, we show that the pessimistic target in PBRL aligns closely with the recent theoretical investigation on offline RL (Jin et al., 2021; Xie et al., 2021a). From the theoretical perspective, an appropriate uncertainty quantification is essential to the provable efficiency in offline RL. Specifically, the $\xi$-uncertainty quantifier plays a central role in the analysis of both online and offline RL (Jaksch et al., 2010; Azar et al., 2017; Wang et al., 2020a; Jin et al., 2020; 2021; Xie et al., 2021a;b).

**Definition 1** ($\xi$-Uncertainty Quantifier (Jin et al., 2021)). *The set of penalization $\{\Gamma_t\}_{t \in [T]}$ forms a $\xi$-Uncertainty Quantifier if it holds with probability at least $1 - \xi$ that*

$$|\widehat{\mathcal{T}}V_{t+1}(s,a) - \mathcal{T}V_{t+1}(s,a)| \leq \Gamma_t(s,a)$$

*for all $(s,a) \in \mathcal{S} \times \mathcal{A}$, where $\mathcal{T}$ is the Bellman equation and $\widehat{\mathcal{T}}$ is the empirical Bellman equation that estimates $\mathcal{T}$ based on the offline data.*

In linear MDPs (Jin et al., 2020; Wang et al., 2020a; Jin et al., 2021) where the transition kernel and reward function are assumed to be linear to the state-action representation $\phi : \mathcal{S} \times \mathcal{A} \to \mathbb{R}^d$, The following LCB-penalty (Abbasi-Yadkori et al., 2011; Jin et al., 2020) is known to be a $\xi$-uncertainty quantifier for appropriately selected $\{\beta_t\}_{t \in [T]}$,

$$\Gamma^{\text{lcb}}(s_t, a_t) = \beta_t \cdot \big[ \phi(s_t, a_t)^{\top} \Lambda_t^{-1} \phi(s_t, a_t) \big]^{1/2}, \qquad (8)$$

where $\Lambda_t = \sum_{i=1}^m \phi(s_t^i, a_t^i) \phi(s_t^i, a_t^i)^{\top} + \lambda \cdot \mathbf{I}$ accumulates the features of state-action pairs in $\mathcal{D}_{\text{in}}$ and plays the role of a pseudo-count intuitively. We remark that under such linear MDP assumptions, the penalty proposed in PBRL and $\Gamma^{\text{lcb}}(s_t, a_t)$ in linear MDPs is equivalent under a Bayesian perspective. Specifically, we make the following claim.

**Claim 1.** *In linear MDPs, the proposed bootstrapped uncertainty $\beta_t \cdot \mathcal{U}(s_t, a_t)$ is an estimation to the LCB-penalty $\Gamma^{\text{lcb}}(s_t, a_t)$ in Eq. (8) for an appropriately selected tuning parameter $\beta_t$.*

We refer to §A for a detailed explanation and proof. Intuitively, the bootstrapped $Q$-functions estimates a non-parametric $Q$-posterior (Osband et al., 2016; 2018a). Correspondingly, the uncertainty quantifier $\mathcal{U}(s_t, a_t)$ estimates the standard deviation of the $Q$-posterior, which scales with the LCB-penalty in linear MDPs. As an example, we show that under the tabular setting, $\Gamma^{\text{lcb}}(s_t, a_t)$ is approximately proportional to the reciprocal pseudo-count of the corresponding state-action pair in the dataset (See Lemma 2 in §A). In offline RL, such uncertainty quantification measures how trustworthy the value estimations on state-action pairs are. A low LCB-penalty (or high pseudo-count) indicates that the corresponding state-action pair aligns with the support of offline data.

Under the linear MDP or Bellman-consistent assumptions, penalizing the estimated value function based on the uncertainty quantification is known to yield an efficient offline RL algorithm (Jin et al., 2021; Xie et al., 2021a;b). However, due to the large extrapolation error of neural networks, we find that solely penalizing the value function of the in-distribution samples is insufficient to regularize the fitted value functions of OOD state-action pairs.

A key to the success of linear MDP algorithms (Jin et al., 2020; Wang et al., 2020a; Jin et al., 2021) is the extrapolation ability through $L_2$-regularization in the least-squares value iteration (LSVI), which guarantees that the linear parameterized value functions behave reasonably on OOD state-action pairs. From a Bayesian perspective, such $L_2$-regularization enforces a Gaussian prior on the estimated parameter of linear approximations, which regularizes the value function estimation on OOD state-action pairs with limited data available. Nevertheless, our empirical study in §D shows that $L_2$-regularization is not sufficient to regularize the OOD behavior of neural networks.

To this end, PBRL introduces a direct regularization over an OOD dataset. From the theoretical perspective, we observe that adding OOD datapoint $(s^{\text{ood}}, a^{\text{ood}}, y)$ into the offline dataset leads to an equivalent regularization to the $L_2$-regularization under the linear MDP assumption. Specifically, in linear MDPs, such additional OOD sampling yields a covariate matrix of the following form,

$$\widetilde{\Lambda} = \sum\nolimits_{i=1}^m \phi(s_t^i, a_t^i)\phi(s_t^i, a_t^i)^\top + \sum\nolimits_{(s^{\text{ood}}, a^{\text{ood}}, y) \in \mathcal{D}_{\text{ood}}} \phi(s^{\text{ood}}, a^{\text{ood}})\phi(s^{\text{ood}}, a^{\text{ood}})^\top, \quad (9)$$

where the matrix $\Lambda_{\text{ood}} = \sum_{(s^{\text{ood}}, a^{\text{ood}}, y) \in \mathcal{D}_{\text{ood}}} \phi(s^{\text{ood}}, a^{\text{ood}})\phi(s^{\text{ood}}, a^{\text{ood}})^\top$ plays the role of the $\lambda \cdot \mathbf{I}$ prior in LSVI. It remains to design a proper target $y$ in the OOD datapoint $(s^{\text{ood}}, a^{\text{ood}}, y)$. The following theorem show that setting $y = \mathcal{T}V_{h+1}(s^{\text{ood}}, a^{\text{ood}})$ leads to a valid $\xi$-uncertainty quantifier under the linear MDP assumption.

**Theorem 1.** *Let $\Lambda_{\text{ood}} \succeq \lambda \cdot I$. Under the linear MDP assumption, for all the OOD datapoint $(s^{\text{ood}}, a^{\text{ood}}, y) \in \mathcal{D}_{\text{ood}}$, if we set $y = \mathcal{T}V_{t+1}(s^{\text{ood}}, a^{\text{ood}})$, it then holds for $\beta_t = \mathcal{O}\big(T \cdot \sqrt{d} \cdot log(T/\xi)\big)$ that $\Gamma_t^{\text{lcb}}(s_t, a_t) = \beta_t \big[\phi(s_t, a_t)^\top \Lambda_t^{-1} \phi(s_t, a_t)\big]^{1/2}$ forms a valid $\xi$-uncertainty quantifier.*

We refer to §A for a detailed discussion and proof. Theorem 1 shows that if we set $y = \mathcal{T}V_{t+1}(s^{\text{ood}}, a^{\text{ood}})$, the bootstrapped uncertainty based on disagreement among ensembles is a valid $\xi$-uncertainty quantifier. However, such an OOD target is impossible to obtain in practice as it requires knowing the transition at the OOD datapoint $(s^{\text{ood}}, a^{\text{ood}})$. In practice, if TD error is sufficiently minimized, then $Q(s^{\text{ood}}, a^{\text{ood}})$ should well estimate the target $\mathcal{T}V_{t+1}$. Thus, in PBRL, we utilize

$$y = Q(s^{\text{ood}}, a^{\text{ood}}) - \Gamma^{\text{lcb}}(s^{\text{ood}}, a^{\text{ood}}) \quad (10)$$

as the OOD target, where we introduce an additional penalization $\Gamma^{\text{lcb}}(s^{\text{ood}}, a^{\text{ood}})$ to enforce pessimism. In addition, we remark that in theory, we require that the embeddings of the OOD sample are isotropic in the sense that the eigenvalues of the corresponding covariate matrix $\Lambda_{\text{ood}}$ are lower bounded. Such isotropic property can be guaranteed by randomly generating states and actions. In practice, we find that randomly generating states is more expensive than randomly generating actions. Meanwhile, we observe that randomly generating actions alone are sufficient to guarantee reasonable empirical performance since the generated OOD embeddings are sufficiently isotropic. Thus, in our experiments, we randomly generate OOD actions according to our current policy and sample OOD states from the in-distribution dataset.

## 4 RELATED WORKS

Previous model-free offline RL algorithms typically rely on policy constraints to restrict the learned policy from producing the OOD actions. In particular, previous works add behavior cloning (BC) loss in policy training (Fujimoto et al., 2019; Fujimoto & Gu, 2021; Ghasemipour et al., 2021), measure the divergence between the behavior policy and the learned policy (Kumar et al., 2019; Wu et al., 2019; Kostrikov et al., 2021), apply advantage-weighted constraints to balance BC and advantages (Siegel et al., 2020; Wang et al., 2020b), penalize the prediction-error of a variational auto-encoder (Rezaeifar et al., 2021), and learn latent actions (or primitives) from the offline data (Zhou et al., 2020; Ajay et al., 2021). Nevertheless, such methods may cause overly conservative value functions and are easily affected by the behavior policy (Nair et al., 2020; Lee et al., 2021b). We remark that the OOD actions that align closely with the support of offline data could also be trustworthy. CQL (Kumar et al., 2020) directly minimizes the $Q$-values of OOD samples and thus casts an implicit policy constraint. Our method is related to CQL in the sense that both PBRL and CQL enforce conservatism in $Q$-learning. In contrast, we conduct explicit uncertainty quantification for OOD actions, while CQL penalizes the $Q$-values of all OOD samples equally.

In contrast with model-free algorithms, model-based algorithms learn the dynamics model directly with supervised learning. Similar to our work, MOPO (Yu et al., 2020) and MOReL (Kidambi

et al., 2020) incorporate ensembles of dynamics-models for uncertainty quantification, and penalize the value function through pessimistic updates. Other than the uncertainty quantification, previous model-based methods also attempt to constrain the learned policy through BC loss (Matsushima et al., 2020), advantage-weighted prior (Cang et al., 2021), CQL-style penalty (Yu et al., 2021), and Riemannian submanifold (Tennenholtz et al., 2021). Decision Transformer (Chen et al., 2021) builds a transformer-style dynamic model and casts the problem of offline RL as conditional sequence modeling. However, such model-based methods may suffer from additional computation costs and may perform suboptimally in complex environments (Chua et al., 2018; Janner et al., 2019). In contrast, PBRL conducts model-free learning and is less affected by such challenges.

Our method is related to the previous online RL exploration algorithms based on uncertainty quantification, including bootstrapped $Q$-networks (Bai et al., 2021; Lee et al., 2021a), ensemble dynamics (Sekar et al., 2020), Bayesian NN (O'Donoghue et al., 2018; Azizzadenesheli et al., 2018), and distributional value functions (Mavrin et al., 2019; Nikolov et al., 2019). Uncertainty quantification is more challenging in offline RL than its online counterpart due to the limited coverage of offline data and the distribution shift of the learned policy. In model-based offline RL, MOPO (Yu et al., 2020) and MOReL (Kidambi et al., 2020) incorporate ensemble dynamics-model for uncertainty quantification. BOPAH (Lee et al., 2020) combines uncertainty penalization and behavior-policy constraints. In model-free offline RL, UWAC (Wu et al., 2021) adopts dropout-based uncertainty (Gal & Ghahramani, 2016) while relying on policy constraints in learning value functions. In contrast, PBRL does not require additional policy constraints. In addition, according to the study in image prediction with data shift (Ovadia et al., 2019), the bootstrapped uncertainty is more robust to data shift than the dropout-based approach. EDAC (An et al., 2021) is a concurrent work that uses the ensemble $Q$-network. Specifically, EDAC calculates the gradients of each Q-function and diversifies such gradients to ensure sufficient penalization for OOD actions. In contrast, PBRL penalizes the OOD actions through direct OOD sampling and the associated uncertainty quantification.

Our algorithm is inspired by the recent advances in the theory of both online RL and offline RL. Previous works propose provably efficient RL algorithms under the linear MDP assumption for both the online setting (Jin et al., 2020) and offline setting (Jin et al., 2021), which we follow for our analysis. In addition, previous works also study the offline RL under the Bellman completeness assumptions (Modi et al., 2021; Uehara et al., 2021; Xie et al., 2021a; Zanette et al., 2021) and the model-based RL under the kernelized nonlinear regulator (KNR) setting (Kakade et al., 2020; Mania et al., 2020; Chang et al., 2021). In contrast, our paper focus on model-free RL.

## 5 EXPERIMENTS

In experiments, we include an additional algorithm named *PBRL-prior*, which is a slight modification of PBRL by incorporating random prior functions (Osband et al., 2018b). The random prior technique is originally proposed for online exploration in Bootstrapped DQN (Osband et al., 2016). Specifically, each $Q$-function in *PBRL-Prior* contains a trainable network $Q_\theta^k$ and a prior network $p_k$, where $p_k$ is randomly initialized and is fixed in training. The prediction of each $Q$-function is the sum of the trainable network and the fixed prior, $Q_p^k = Q_\theta^k + p_k$, where $Q_\theta^k$ and $p_k$ shares the same network architecture. The random prior function increases the diversity among ensemble members and improves the generalization of bootstrapped functions (Osband et al., 2018b). We adopt SAC (Haarnoja et al., 2018) as the basic actor-critic architecture for both *PBRL* and *PBRL-prior*. We refer to §B for the implementation details. The code is available at https://github.com/Baichenjia/PBRL.

In D4RL benchmark (Fu et al., 2020) with various continuous-control tasks and datasets, we compare the baseline algorithms on the Gym and Adroit domains, which are more extensively studied in the previous research. We compare *PBRL* and *PBRL-Prior* with several state-of-the-art algorithms, including (i) *BEAR* (Kumar et al., 2019) that enforces policy constraints through the MMD distance, (ii) *UWAC* (Wu et al., 2021) that improves *BEAR* through dropout uncertainty-weighted update, (iii) *CQL* (Kumar et al., 2020) that learns conservative value functions by minimizing $Q$-values of OOD actions, (iv) *MOPO* (Yu et al., 2020) that quantifies the uncertainty through ensemble dynamics in a model-based setting, and (v) *TD3-BC* (Fujimoto & Gu, 2021), which adopts adaptive BC constraint to regularize the policy in training.

| | | BEAR | UWAC | CQL | MOPO | TD3-BC | **PBRL** | **PBRL-Prior** |
|---|---|---|---|---|---|---|---|---|
| Random | HalfCheetah | 2.3 ±0.0 | 2.3 ±0.0 | 17.5 ±1.5 | 35.9 ±2.9 | 11.0 ±1.1 | 11.0 ±5.8 | 13.1 ±1.2 |
| | Hopper | 3.9 ±2.3 | 2.7 ±0.3 | 7.9 ±0.4 | 16.7 ±12.2 | 8.5 ±0.6 | 26.8 ±9.3 | 31.6 ±0.3 |
| | Walker2d | 12.8 ±10.2 | 2.0 ±0.4 | 5.1 ±1.3 | 4.2 ±5.7 | 1.6 ±1.7 | 8.1 ±4.4 | 8.8 ±6.3 |
| Medium | HalfCheetah | 43.0 ±0.2 | 42.2 ±0.4 | 47.0 ±0.5 | 73.1 ±2.4 | 48.3 ±0.3 | 57.9 ±1.5 | 58.2 ±1.5 |
| | Hopper | 51.8 ±4.0 | 50.9 ±4.4 | 53.0 ±28.5 | 38.3 ±34.9 | 59.3 ±4.2 | 75.3 ±31.2 | 81.6 ±14.5 |
| | Walker2d | -0.2 ±0.1 | 75.4 ±3.0 | 73.3 ±17.7 | 41.2 ±30.8 | 83.7 ±2.1 | 89.6 ±0.7 | 90.3 ±1.2 |
| Medium Replay | HalfCheetah | 36.3 ±3.1 | 35.9 ±3.7 | 45.5 ±0.7 | 69.2 ±1.1 | 44.6 ±0.5 | 45.1 ±8.0 | 49.5 ±0.8 |
| | Hopper | 52.2 ±19.3 | 25.3 ±1.7 | 88.7 ±12.9 | 32.7 ±9.4 | 60.9 ±18.8 | 100.6 ±1.0 | 100.7 ±0.4 |
| | Walker2d | 7.0 ±7.8 | 23.6 ±6.9 | 81.8 ±2.7 | 73.7 ±9.4 | 81.8 ±5.5 | 77.7 ±14.5 | 86.2 ±3.4 |
| Medium Expert | HalfCheetah | 46.0 ±4.7 | 42.7 ±0.3 | 75.6 ±25.7 | 70.3 ±21.9 | 90.7 ±4.3 | 92.3 ±1.1 | 93.6 ±2.3 |
| | Hopper | 50.6 ±25.3 | 44.9 ±8.1 | 105.6 ±12.9 | 60.6 ±32.5 | 98.0 ±9.4 | 110.8 ±0.8 | 111.2 ±0.7 |
| | Walker2d | 22.1 ±44.9 | 96.5 ±9.1 | 107.9 ±1.6 | 77.4 ±27.9 | 110.1 ±0.5 | 110.1 ±0.3 | 109.8 ±0.2 |
| Expert | HalfCheetah | 92.7 ±0.6 | 92.9 ±0.6 | 96.3 ±1.3 | 81.3 ±21.8 | 96.7 ±1.1 | 92.4 ±1.7 | 96.2 ±2.3 |
| | Hopper | 54.6 ±21.0 | 110.5 ±0.5 | 96.5 ±28.0 | 62.5 ±29.0 | 107.8 ±7 | 110.5 ±0.4 | 110.4 ±0.3 |
| | Walker2d | 106.6 ±6.8 | 108.4 ±0.4 | 108.5 ±0.5 | 62.4 ±3.2 | 110.2 ±0.3 | 108.3 ±0.3 | 108.8 ±0.2 |
| | **Average** | 38.78 ±10.0 | 50.41 ±2.7 | 67.35 ±9.1 | 53.3 ±16.3 | 67.55 ±3.8 | 74.37 ±5.3 | 76.66 ±2.4 |

Table 1: Average normalized score and the standard deviation of all algorithms over five seeds in Gym. The highest performing scores are highlighted. The score of TD3-BC is the reported scores in Table 7 of Fujimoto & Gu (2021). The scores for other baselines are obtained by re-training with the 'v2' dataset of D4RL (Fu et al., 2020).

**Results in Gym domain.** The Gym domain includes three environments (HalfCheetah, Hopper, and Walker2d) with five dataset types (random, medium, medium-replay, medium-expert, and expert), leading to a total of 15 problem setups. We train all the baseline algorithms in the latest released 'v2' version dataset, which is also adopted in TD3-BC (Fujimoto & Gu, 2021) for evaluation. For methods that are originally evaluated on the 'v0' dataset, we retrain with their respective official implementations on the 'v2' dataset. We refer to §B for the training details. We train each method for one million time steps and report the final evaluation performance through online interaction. Table 1 reports the normalized score for each task and the corresponding average performance. We find CQL and TD3-BC perform the best among all baselines, and PBRL outperforms the baselines in most of the tasks. In addition, PBRL-Prior slightly outperforms PBRL and is more stable in training with a reduced variance among different seeds.

We observe that compared with the baseline algorithms, PBRL has strong advantages in the non-optimal datasets, including medium, medium-replay, and medium-expert. In addition, compared with the policy-constraint baselines, PBRL exploits the optimal trajectory covered in the dataset in a theoretically grounded way and is less affected by the behavior policy. We report the average training curves in Fig. 2. In addition, we remark that the performance of PBRL and PBRL-Prior are weaker than TD3-BC and CQL in the early stage of training, indicating that the uncertainty quantification is inaccurate initially. Nevertheless, PBRL and PBRL-Prior converge to better policies that outperform the baselines in the learning of uncertainty quantifiers, demonstrating the effectiveness of uncertainty penalization and OOD sampling.

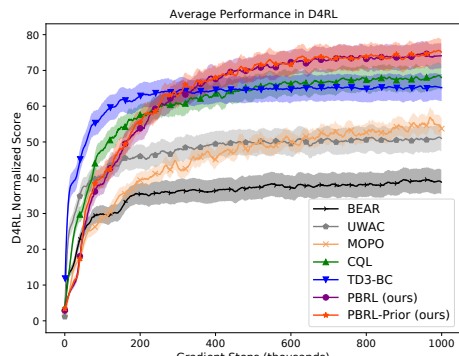

Figure 2: Average training curve in Gym.

**Results in Adroit domain.** The adroit tasks are more challenging than the Gym domain in task complexity. In addition, the use of human demonstration in the dataset makes the task even more challenging in the offline setting. We defer the results to §E. We observe that CQL and BC have the best average performance in all baselines, and PBRL outperforms baselines in most of the tasks.

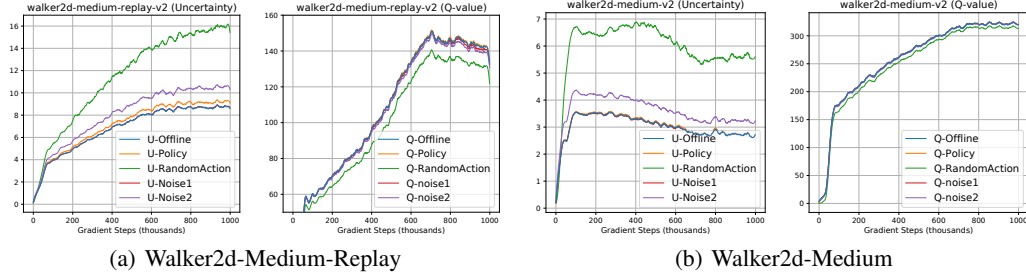

(a) Walker2d-Medium-Replay          (b) Walker2d-Medium

Figure 3: The uncertainty and $Q$-value for different state-action sets in the training process.

**Uncertainty quantification.** To verify the effectiveness of the bootstrapped uncertainty quantification, we record the uncertainty quantification for different sets of state-action pairs in training. In our experiments, we consider sets that have the same states from the offline dataset but with different types of actions, including (i) $a_{\text{offline}}$, which is drawn from the offline in-distribution transition; (ii) $a_{\text{policy}}$, which is selected by the training policy; (iii) $a_{\text{rand}}$, which is uniformly sampled from the action space of the corresponding task; (iv) $a_{\text{noise1}} = a_{\text{offline}} + \mathcal{N}(0, 0.1)$, which adds a small Gaussian noise onto the offline action to represent state-action pair that is close to in-distribution data; and (v) $a_{\text{noise2}} = a_{\text{offline}} + \mathcal{N}(0, 0.5)$, which adds a large noise to represent the OOD action.

We compute the uncertainty and $Q$-value in 'Walker2d' task with two datasets ('medium-replay' and 'medium'). The results are shown in Fig. 3. We observe that (i) PBRL yields large uncertainties for $(s, a_{\text{noise2}})$ and $(s, a_{\text{random}})$, indicating that the uncertainty quantification is high on OOD samples. (ii) The $(s, a_{\text{offline}})$ pair has the smallest uncertainty in both settings as it is an in-distribution sample. (iii) The $(s, a_{\text{noise1}})$ pair has slightly higher uncertainty compared to $(s, a_{\text{offline}})$, showing that the uncertainty quantification rises smoothly from the in-distribution actions to the OOD actions. (iv) The $Q$-function of the learned policy $(s, a_{\text{policy}})$ is reasonable and does not deviate much from the in-distribution actions, which shows that the learned policy does not take actions with high uncertainty due to the penalty with uncertainty quantification. In addition, we observe that there is no superior maximum for OOD actions, indicating that by incorporating the uncertainty quantification and OOD sampling, the $Q$-functions obtained by PBRL does not suffer from the extrapolation error.

**Ablation study.** In the following, we briefly report the result of the ablation study. We refer to §C and §D for the details. (i) *Number of bootstrapped-Q*. We attempt different numbers $K$ of bootstrapped-$Q$ in PBRL, and find the performance to be reasonable for $K \geq 6$. (ii) *Penalization in $\widehat{\mathcal{T}}^{\text{in}}$*. We conduct experiments with the penalized reward discussed in §3.2 and find the penalized reward does not improve the performance. (iii) *Factor $\beta_{\text{in}}$ in $\widehat{\mathcal{T}}^{\text{in}}$*. We conduct experiments with different $\beta_{\text{in}}$ from $\{0.1, 0.01, 0.001, 0.0001\}$ to study the sensitiveness. Our experiments show that PBRL performs the best for $\beta_{\text{in}} \in [0.0001, 0.01]$. (iv) *Factor $\beta_{\text{ood}}$*. We use a large $\beta_{\text{ood}}$ at the initial training stage to enforce strong regularization over the OOD actions, and gradually decrease $\beta_{\text{ood}}$ in training. We conduct experiments by setting $\beta_{\text{ood}}$ to different constants, and find our decaying strategy generalizes better among tasks. (v) *The learning target of OOD actions*. We change the learning target of OOD actions to the most pessimistic zero target $y = 0$ and find such a setting leads to overly pessimistic value functions with suboptimal performance. (vi) *Regularization types*. We conduct experiments with different regularization types, including our proposed OOD sampling, $L_2$-regularization, spectral normalization, pessimistic initialization, and no regularization. We find OOD sampling the only reasonable regularization strategy and defer the complete report to §D.

## 6 CONCLUSION

In this paper, we propose PBRL, an uncertainty-based model-free offline RL algorithm. We propose bootstrapped uncertainty to guide the provably efficient pessimism, and a novel OOD sampling technique to regularize the OOD actions. PBRL is closely related to the provable efficient offline RL algorithm under the linear MDP assumption. Our experiments show that PBRL outperforms several strong offline RL baselines in the D4RL environments. PBRL exploits the optimal trajectories contained in the suboptimal dataset and is less affected by the behavior policy. Meanwhile, we show that PBRL produces reliable uncertainty quantifications incorporated with OOD sampling.

## REPRODUCIBILITY STATEMENT

The code of our work is available at `https://github.com/Baichenjia/PBRL`. The environments, datasets, and hyper-parameters of our experiments are are given in the appendix. All settings, assumptions, lemmas, and theorems are proved and are discussed in detail in our appendix.

## ACKNOWLEDGEMENTS

This work was supported in part by the National Natural Science Foundation of China under Grant 51935005, in part by the Fundamental Research Program under Grant JCKY20200603C010. The authors thank the anonymous reviewers, whose invaluable comments and suggestions have helped us to improve the paper.

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

## A  THEORETICAL PROOF

### A.1  BACKGROUND OF LCB-PENALTY IN LINEAR MDPs

In this section, we introduce the provably efficient LCB-penalty in linear MDPs (Abbasi-Yadkori et al., 2011; Jin et al., 2020; 2021). We consider the setting of $\gamma = 1$ in the following. In linear MDPs, the feature map of the state-action pair takes the form of $\phi : \mathcal{S} \times \mathcal{A} \to \mathbb{R}^d$, and the transition kernel and reward function are assumed to be linear in $\phi$. As a result, for any policy $\pi$, the state-action value function is also linear in $\phi$ (Jin et al., 2020), that is,

$$Q^\pi(s_t^i, a_t^i) = \widehat{w}_t^\top \phi(s_t^i, a_t^i). \tag{11}$$

The parameter $w_t$ can be solved in the closed-form by following the Least-Squares Value Iteration (LSVI) algorithm, which minimizes the following loss function,

$$\widehat{w}_t = \min_{w \in \mathbb{R}^d} \sum_{i=1}^m \big(\phi(s_t^i, a_t^i)^\top w - r(s_t^i, a_t^i) - V_{t+1}(s_{t+1}^i)\big)^2 + \lambda \cdot \|w\|_2^2, \tag{12}$$

where $V_{t+1}$ is the estimated value function in the $(t+1)$-th step, and $r(s_t^i, a_t^i) + V_{t+1}(s_{t+1}^i)$ is the target of LSVI. The explicit solution to (12) takes the form of

$$\widehat{w}_t = \Lambda_t^{-1} \sum_{i=1}^m \phi(s_t^i, a_t^i)\big(V_{t+1}(s_{t+1}^i) + r(s_t^i, a_t^i)\big), \quad \Lambda_t = \sum_{i=1}^m \phi(s_t^i, a_t^i)\phi(s_t^i, a_t^i)^\top + \lambda \cdot \mathbf{I}. \tag{13}$$

Here $\Lambda_t$ accumulate the state-action features from the training buffer. Based on the solution of $w_t$, the action-value function can be estimated by $Q_t(s_t, a_t) \approx \widehat{w}_t^\top \phi(s_t, a_t)$. In addition, in offline RL with linear function assumption, the following LCB-penalty yields an uncertainty quantification,

$$\Gamma^{\mathrm{lcb}}(s_t, a_t) = \beta_t \cdot \big[\phi(s_t, a_t)^\top \Lambda_t^{-1} \phi(s_t, a_t)\big]^{1/2}, \tag{14}$$

which measures the confidence interval of the $Q$-functions with the given training data (Abbasi-Yadkori et al., 2011; Jin et al., 2020; 2021). In offline RL, the pessimistic value function $\widehat{Q}_t(s_t, a_t)$ penalizes $Q_t$ by the uncertainty quantification $\Gamma^{\mathrm{lcb}}(s_t, a_t)$ as a penalty as follows,

$$\begin{aligned} \widehat{Q}_t(s_t, a_t) &= Q_t(s_t, a_t) - \Gamma^{\mathrm{lcb}}(s_t, a_t) \\ &= w_t^\top \phi(s_t, a_t) - \Gamma^{\mathrm{lcb}}(s_t, a_t), \end{aligned} \tag{15}$$

where $w_t$ is defined in Eq. (12). Under the linear MDP setting, such pessimistic value iteration is known to be information-theoretically optimal (Jin et al., 2021). In addition, exploration with $\Gamma^{\mathrm{lcb}}(s_t, a_t)$ as a bonus is also provably efficient in the online RL setting (Abbasi-Yadkori et al., 2011; Jin et al., 2020).

### A.2  CONNECTION BETWEEN THE BOOTSTRAPPED UNCERTAINTY AND $\Gamma^{\mathrm{lcb}}$

In the sequel, we consider a Bayesian linear regression perspective of LSVI in Eq. (12). According to the Bellman equation, the objective of LSVI is to approximate the Bellman target $b_t^i = r(s_t^i, a_t^i) + V_{t+1}(s_{t+1}^i)$ with the $Q$-function $Q_t$, where $V_{t+1}$ is the estimated value function in the $(t+1)$-th step. In linear MDPs, We parameterize the $Q$-function by $Q_t(s_t, a_t) = \widehat{w}_t^\top \phi(s_t, a_t)$. We further define the noise $\epsilon$ in this least-square problem as follows,

$$\epsilon = b_t^i - w_t^\top \phi(s_t, a_t), \tag{16}$$

where $w_t$ is the underlying true parameter. In the offline dataset with $\mathcal{D}_{\mathrm{in}} = \{(s_t^i, a_t^i, s_{t+1}^i)\}_{i \in [m]}$, we denote by $\widehat{w}_t$ the Bayesian posterior of $w$ given the dataset $\mathcal{D}_{\mathrm{in}}$. In addition, we assume that we are given a Gaussian prior of the parameter $w \sim \mathcal{N}(0, \mathbf{I}/\lambda)$ as a non-informative prior. The following Lemma establishes connections between bootstrapped uncertainty and the LCB-penalty $\Gamma^{\mathrm{lcb}}$.

**Lemma 1** (Formal Version of Claim 1). *We assume that $\epsilon$ follows the standard Gaussian distribution $\mathcal{N}(0, 1)$ given the state-action pair $(s_t^i, a_t^i)$. It then holds for the posterior $w_t$ given $\mathcal{D}_{\mathrm{in}}$ that*

$$\mathrm{Var}_{\widehat{w}_t}\big(Q_t(s_t^i, a_t^i)\big) = \mathrm{Var}_{\widehat{w}_t}\big(\phi(s_t^i, a_t^i)^\top \widehat{w}_t\big) = \phi(s_t^i, a_t^i)^\top \Lambda_t^{-1} \phi(s_t^i, a_t^i), \quad \forall(s_t^i, a_t^i) \in \mathcal{S} \times \mathcal{A}. \tag{17}$$

*Proof.* The proof follows the standard analysis of Bayesian linear regression. Under the assumption that $\epsilon \sim \mathcal{N}(0, 1)$, we obtain that

$$b_t^i \mid (s_t^i, a_t^i), \widehat{w} \sim \mathcal{N}\big(\widehat{w}_t^\top \phi(s_t^i, a_t^i), 1\big). \tag{18}$$

Recall that we have the prior distribution $w \sim \mathcal{N}(0, \mathbf{I}/\lambda)$. Our objective is to compute the posterior density $\widehat{w}_t = w \mid \mathcal{D}_{\text{in}}$. It holds from Bayes rule that

$$\log p(\widehat{w} \mid \mathcal{D}_{\text{in}}) = \log p(\widehat{w}) + \log p(\mathcal{D}_{\text{in}} \mid \widehat{w}) + \text{Const.}, \tag{19}$$

where $p(\cdot)$ denote the probability density function of the respective distributions. Plugging (18) and the probability density function of Gaussian distribution into (19) yields

$$\log p(\widehat{w} \mid \mathcal{D}_{\text{in}}) = -\|\widehat{w}\|^2/2 - \sum_{i=1}^m \|\widehat{w}^\top \phi(s_t^i, a_t^i) - y_t^i\|^2/2 + \text{Const.}$$

$$= -(\widehat{w} - \mu_t)^\top \Lambda_t^{-1}(\widehat{w} - \mu_t)/2 + \text{Const.},$$

where we define

$$\mu_t = \Lambda_t^{-1} \sum_{i=1}^m \phi(s_t^i, a_t^i) y_t^i, \qquad \Lambda_t = \sum_{i=1}^m \phi(s_t^i, a_t^i)\phi(s_t^i, a_t^i)^\top + \lambda \cdot \mathbf{I}. \tag{20}$$

Then we obtain that $\widehat{w}_t = w \mid \mathcal{D}_{\text{in}} \sim \mathcal{N}(\mu_t, \Lambda_t^{-1})$. It then holds for all $(s_t, a_t) \in \mathcal{S} \times \mathcal{A}$ that

$$\text{Var}\big(\phi(s_t^i, a_t^i)^\top \widehat{w}_t\big) = \text{Var}\big(Q_t(s_t^i, a_t^i)\big) = \phi(s_t^i, a_t^i)^\top \Lambda_t^{-1} \phi(s_t^i, a_t^i), \tag{21}$$

which concludes the proof. $\qquad\square$

In Lemma 1, we show that the standard deviation of the $Q$-posterior is equivalent to the LCB-penalty $\text{Var}\big(Q(s_t, a_t)\big) = \phi(s_t, a_t)^\top \Lambda_t^{-1} \phi(s_t, a_t)$ introduced in §A.1. Recall that our proposed bootstrapped uncertainty takes the form of

$$\mathcal{U}(s_t, a_t) \approx \text{Std}\big(Q^k(s_t, a_t)\big), \tag{22}$$

which is the standard deviation of the bootstrapped $Q$-functions. Such bootstrapping serves as an estimation of the posterior of $Q$-functions (Osband et al., 2016). Thus, our proposed uncertainty quantification can be seen as an estimation of the LCB-penalty under the linear MDP assumptions.

In addition, under the tabular setting where the states and actions are finite, the LCB-penalty takes a simpler form, which we show in the following lemma.

**Lemma 2.** *In tabular MDPs, the bootstrapped uncertainty $\mathcal{U}(s, a)$ is approximately proportional to the reciprocal-count of $(s, a)$, that is,*

$$\mathcal{U}(s, a) \approx \Gamma^{\text{lcb}}(s, a)/\beta_t = \frac{1}{\sqrt{N_{s,a} + \lambda}}. \tag{23}$$

*Proof.* In tabular MDPs, we consider the joint state-action space $d = |\mathcal{S}| \times |\mathcal{A}|$. Then $j$-th state-action pair can be encoded as a one-hot vector as $\phi(s, a) \in \mathbb{R}^d$, where $j \in [0, d-1]$, thus is a special case of the linear MDP (Yang & Wang, 2019; Jin et al., 2020). Specifically, we define

$$\phi(s_j, a_j) = \begin{bmatrix} 0 \\ \vdots \\ 1 \\ \vdots \\ 0 \end{bmatrix} \in \mathbb{R}^d, \qquad \phi(s_j, a_j)\phi(s_j, a_j)^\top = \begin{bmatrix} 0 & \cdots & 0 & \cdots & 0 \\ \vdots & \ddots & & & \vdots \\ 0 & & 1 & & 0 \\ \vdots & & & \ddots & \vdots \\ 0 & \cdots & 0 & \cdots & 0 \end{bmatrix} \in \mathbb{R}^{d \times d}, \tag{24}$$

where the value of $\phi(s_j, a_j)$ is 1 at the $j$-th entry and 0 elsewhere. Then the matrix $\Lambda_j = \sum_{i=0}^m \phi(s_j^i, a_j^i)\phi(s_j^i, a_j^i)^\top + \lambda \cdot \mathbf{I}$ is the sum of $\phi(s_j, a_j)\phi(s_j, a_j)^\top$ over $(s_j, a_j) \in \mathcal{D}_{\text{in}}$, which takes the form of

$$\Lambda_j = \begin{bmatrix} n_0+\lambda & 0 & & \cdots & & 0 \\ 0 & n_1+\lambda & & \cdots & & 0 \\ \vdots & & \ddots & & & \vdots \\ 0 & \cdots & & n_j+\lambda & & 0 \\ \vdots & & & & \ddots & \vdots \\ 0 & \cdots & & \cdots & & n_{d-1}+\lambda \end{bmatrix}, \tag{25}$$

where the $j$-th diagonal element of $\Lambda_j$ is the corresponding counts for state-action $(s_j, a_j)$, i.e.,

$$n_j = N_{s_j, a_j}.$$

It thus holds that

$$\left[ \phi(s_j, a_j)^\top \Lambda_j^{-1} \phi(s_j, a_j) \right]^{1/2} = \frac{1}{\sqrt{N_{s_j, a_j} + \lambda}}, \tag{26}$$

which concludes the proof. $\qquad\square$

### A.3 REGULARIZATION WITH OOD SAMPLING

In this section, we discuss how OOD sampling plays the role of regularization in RL, which regularizes the extrapolation behavior of the estimated $Q$-functions on OOD samples.

Similar to §A.2, we consider the setup of LSVI-UCB (Jin et al., 2020) under linear MDPs. Specifically, we assume that the transition dynamics and reward function takes the form of

$$\mathbb{P}_t(s_{t+1} \,|\, s_t, a_t) = \langle \psi(s_{t+1}), \phi(s_t, a_t) \rangle, \quad r(s_t, a_t) = \theta^\top \phi(s_t, a_t), \quad \forall (s_{t+1}, a_t, s_t) \in \mathcal{S} \times \mathcal{A} \times \mathcal{S}, \tag{27}$$

where the feature embedding $\phi : \mathcal{S} \times \mathcal{A} \mapsto \mathbb{R}^d$ is known. We further assume that the reward function $r : \mathcal{S} \times \mathcal{A} \mapsto [0, 1]$ is bounded and the feature is bounded by $\|\phi\|_2 \leq 1$. Given the dataset $\mathcal{D}_{\text{in}}$, LSVI iteratively minimizes the least-square loss in Eq. (12). Recall that the explicit solution to Eq. (12) takes the form of

$$\widehat{w}_t = \Lambda_t^{-1} \sum_{i=1}^m \phi(s_t^i, a_t^i)\big(V_{t+1}(s_{t+1}^i) + r(s_t^i, a_t^i)\big), \quad \Lambda_t = \sum_{i=1}^m \phi(s_t^i, a_t^i)\phi(s_t^i, a_t^i)^\top + \lambda \cdot \mathbf{I}. \tag{28}$$

We remark that for the regression problem in Eq. (12), the $L_2$-regularizer $\lambda \cdot \|w\|_2^2$ enforces a Gaussian prior under the notion of Bayesian regression. Such regularization ensures that the linear function approximation $\phi^\top w_t^i$ extrapolates well outside the region covered by the dataset $\mathcal{D}_{\text{in}}$.

However, as shown in §D, we observe that such $L_2$-regularization is ineffective for offline DRL. To this end, we propose *OOD sampling* in our proposed PBRL. To demonstrate the effectiveness of OOD sampling as a regularizer, we consider the following least-squares loss with OOD sampling and without the $L_2$-regularizer,

$$\widetilde{w}_t^i = \min_{w \in \mathbb{R}^d} \sum_{i=1}^m \big(\phi(s_t^i, a_t^i)^\top w - r(s_t^i, a_t^i) - V_{t+1}(s_{t+1}^i)\big)^2 + \sum_{(s^{\text{ood}}, a^{\text{ood}}, y) \in \mathcal{D}_{\text{ood}}} \big(\phi(s^{\text{ood}}, a^{\text{ood}})^\top w - y\big)^2. \tag{29}$$

The explicit solution of Eq. (29) takes the form of

$$\widetilde{w}_t^i = \widetilde{\Lambda}_t^{-1} \bigg( \sum_{i=1}^m \phi(s_t^i, a_t^i)\big(r(s_t^i, a_t^i) + V_{t+1}(s_{t+1}^i)\big) + \sum_{(s^{\text{ood}}, a^{\text{ood}}, y) \in \mathcal{D}_{\text{ood}}} \phi(s^{\text{ood}}, a^{\text{ood}})y \bigg), \tag{30}$$

where

$$\widetilde{\Lambda}_t = \sum_{i=1}^m \phi(s_t^i, a_t^i)\phi(s_t^i, a_t^i)^\top + \sum_{(s^{\text{ood}}, a^{\text{ood}}, y) \in \mathcal{D}_{\text{ood}}} \phi(s^{\text{ood}}, a^{\text{ood}})\phi(s^{\text{ood}}, a^{\text{ood}})^\top. \tag{31}$$

Hence, if we further set $y = 0$ for all $(s^{\text{ood}}, a^{\text{ood}}, y) \in \mathcal{D}_{\text{ood}}$, then (29) enforces a Gaussian prior with the covariance matrix $\Lambda_{\text{ood}}^{-1}$, where we define

$$\Lambda_{\text{ood}} = \sum_{(s^{\text{ood}}, a^{\text{ood}}, y) \in \mathcal{D}_{\text{ood}}} \phi(s^{\text{ood}}, a^{\text{ood}})\phi(s^{\text{ood}}, a^{\text{ood}})^\top. \tag{32}$$

Specifically, if we further enforce $\mathcal{D}_{\text{ood}} = \{(s^j, a^j, 0)\}_{j \in [d]}$ with $\phi(s^j, a^j) = \lambda \cdot e^j$, where $e^j \in \mathbb{R}^d$ is the unit vector with the $j$-th entry equals one, it further holds that $\Lambda_{\text{ood}} = \lambda \cdot \mathbf{I}$ and Eq. (29) is equivalent to Eq. (12). In addition, under the tabular setting, by following the same proof as in Lemma 2, having such OOD samples in the training is equivalent to setting the count in Eq. (26) to be

$$\widetilde{N}_{s_j, a_j} = N_{s_j, a_j} + N_{s_j, a_j}^{\text{ood}},$$

where $N_{s_j,a_j}$ is the occurrence of $(s_j, a_j)$ in the dataset and $N^{\text{ood}}_{s_j,a_j}$ is the occurrence of $(s_j, a_j)$ in the OOD dataset.

However, to enforce such a regularizer without affecting the estimation of value functions, we need to set the target $y$ of the OOD samples to zero. In practice, we find such a setup to be overly pessimistic. Since the $Q$-network is smooth, such a strong regularizer enforces the $Q$-functions to be zero for state-action pairs from both the offline data and OOD data, as show in Fig. 12 and Fig. 13 of §C. We remark that adopting a nonzero OOD target $y$ does not hinder the effect of regularization as it still imposes the same prior in the covariate matrix $\widetilde{\Lambda}_t$. However, adopting nonzero OOD target may introduce additional bias in the value function estimation and the corresponding uncertainty quantification. To maintain a consistent and pessimistic estimation of value functions, one needs to carefully design the nonzero OOD target $y$.

To this end, we recall the definition of a $\xi$-uncertainty quantifier in Definition 1 as follows.

**Definition 2** ($\xi$-Uncertainty Quantifier (Jin et al., 2021))**.** *The set of penalization $\{\Gamma_t\}_{t\in[T]}$ forms a $\xi$-Uncertainty Quantifier if it holds with probability at least $1 - \xi$ that*

$$|\widehat{\mathcal{T}}V_{t+1}(s,a) - \mathcal{T}V_{t+1}(s,a)| \le \Gamma_t(s,a)$$

*for all $(s,a) \in \mathcal{S} \times \mathcal{A}$, where $\mathcal{T}$ is the Bellman operator and $\widehat{\mathcal{T}}$ is the empirical Bellman operator that estimates $\mathcal{T}$ based on the data.*

We remark that here we slightly abuse the notation $\mathcal{T}$ of Bellman operator and write $\mathcal{T}V(s,a) = \mathbb{E}[r(s,a) + V(s') \,|\, s, a]$. Under the linear MDP setup, the empirical estimation $\widehat{\mathcal{T}}V_{t+1}$ is obtained via fitting the least-squares loss in Eq. (29). Thus, the empirical estimation $\widehat{\mathcal{T}}V_{t+1}$ takes the following explicit form,

$$\widehat{\mathcal{T}}V_{t+1}(s_t, a_t) = \phi(s_t, a_t)^\top \widetilde{w}_t,$$

where $\widetilde{w}_t$ is the solution to the least-squares problem defined in Eq. (30). We remark that such $\xi$-uncertainty quantifier plays an important role in the theoretical analysis of RL algorithms, both for online RL and offline RL (Abbasi-Yadkori et al., 2011; Azar et al., 2017; Wang et al., 2020a; Jin et al., 2020; 2021; Xie et al., 2021a;b). Our goal is therefore to design a proper OOD target $y$ such that we can obtain $\xi$-uncertainty quantifier based on the bootstrapped value functions. Our design is motivated by the following theorem.

**Theorem 2.** *Let $\Lambda_{ood} \succeq \lambda \cdot \mathbf{I}$. For all the OOD datapoint $(s^{\text{ood}}, a^{\text{ood}}, y) \in \mathcal{D}_{\text{ood}}$, if we set $y = \mathcal{T}V_{t+1}(s^{\text{ood}}, a^{\text{ood}})$, it then holds for $\beta_t = \mathcal{O}\big(T \cdot \sqrt{d} \cdot \log(T/\xi)\big)$ that*

$$\Gamma_t^{\text{lcb}}(s_t, a_t) = \beta_t \big[\phi(s_t, a_t)^\top \Lambda_t^{-1} \phi(s_t, a_t)\big]^{1/2}$$

*forms a valid $\xi$-uncertainty quantifier.*

*Proof.* Recall that we define the empirical Bellman operator $\widehat{\mathcal{T}}$ as follows,

$$\widehat{\mathcal{T}}V_{t+1}(s_t, a_t) = \phi(s_t, a_t)^\top \widetilde{w}_t,$$

It suffices to upper bound the following difference between the empirical Bellman operator and Bellman operator

$$\mathcal{T}V_{t+1}(s,a) - \widehat{\mathcal{T}}V_{t+1}(s,a) = \phi(s,a)^\top (w_t - \widetilde{w}_t).$$

Here we define $w_t$ as follows

$$w_t = \theta + \int_{\mathcal{S}} V_{t+1}(s_{t+1})\psi(s_{t+1})\mathrm{d}s_{t+1}, \tag{33}$$

where $\theta$ and $\psi$ are defined in Eq. (27). It then holds that

$$\mathcal{T}V_{t+1}(s,a) - \widehat{\mathcal{T}}V_{t+1}(s,a) = \phi(s,a)^\top (w_t - \widetilde{w}_t)$$

$$= \phi(s,a)^\top w_t - \phi(s,a)^\top \widetilde{\Lambda}_t^{-1} \sum_{i=1}^{m} \phi(s_t^i, a_t^i)\big(r(s_t^i, a_t^i) + V_{t+1}^i(s_{t+1}^i)\big)$$

$$- \phi(s,a)^\top \widetilde{\Lambda}_t^{-1} \sum_{(s^{\text{ood}}, a^{\text{ood}}, y) \in \mathcal{D}_{\text{ood}}} \phi(s^{\text{ood}}, a^{\text{ood}})y. \tag{34}$$

where we plug the solution of $\widetilde{w}_t$ in Eq. (30). Meanwhile, by the definitions of $\widetilde{\Lambda}_t$ and $w_t$ in Eq. (31) and Eq. (33), respectively, we have

$$
\phi(s,a)^\top w_t = \phi(s,a)^\top \widetilde{\Lambda}_t^{-1} \widetilde{\Lambda}_t w_t
$$
$$
= \phi(s,a)^\top \widetilde{\Lambda}_t^{-1} \left( \sum_{i=1}^m \phi(s_t^i, a_t^i) \mathcal{T} V_{t+1}(s_t, a_t) + \sum_{(s^{\mathrm{ood}}, a^{\mathrm{ood}}, y) \in \mathcal{D}_{\mathrm{ood}}} \phi(s^{\mathrm{ood}}, a^{\mathrm{ood}}) \mathcal{T} V_{t+1}(s^{\mathrm{ood}}, a^{\mathrm{ood}}) \right).
\tag{35}
$$

Plugging (35) into (34) yields

$$
\mathcal{T} V_{t+1}(s,a) - \widehat{\mathcal{T}} V_{t+1}(s,a) = \text{(i)} + \text{(ii)},
\tag{36}
$$

where we define

$$
\text{(i)} = \phi(s,a)^\top \widetilde{\Lambda}_t^{-1} \sum_{i=1}^m \phi(s_t^i, a_t^i) \big( \mathcal{T} V_{t+1}(s_t^i, a_t^i) - r(s_t^i, a_t^i) - V_{t+1}^i(s_{t+1}^i) \big),
$$
$$
\text{(ii)} = \phi(s,a)^\top \widetilde{\Lambda}_t^{-1} \sum_{(s^{\mathrm{ood}}, a^{\mathrm{ood}}, y) \in \mathcal{D}_{\mathrm{ood}}} \phi(s^{\mathrm{ood}}, a^{\mathrm{ood}}) \big( \mathcal{T} V_{t+1}(s^{\mathrm{ood}}, a^{\mathrm{ood}}) - y \big).
$$

Following the standard analysis based on the concentration of self-normalized process (Abbasi-Yadkori et al., 2011; Azar et al., 2017; Wang et al., 2020a; Jin et al., 2020; 2021) and the fact that $\Lambda_{\mathrm{ood}} \succeq \lambda \cdot I$, it holds that

$$
|\text{(i)}| \le \beta_t \cdot \big[ \phi(s_t, a_t)^\top \Lambda_t^{-1} \phi(s_t, a_t) \big]^{1/2},
\tag{37}
$$

with probability at least $1 - \xi$, where $\beta_t = \mathcal{O}\big( T \cdot \sqrt{d} \cdot \log(T/\xi) \big)$. Meanwhile, by setting $y = \mathcal{T} V_{t+1}(s^{\mathrm{ood}}, a^{\mathrm{ood}})$, it holds that (ii) $= 0$. Thus, we obtain from (36) that

$$
|\mathcal{T} V_{t+1}(s,a) - \widehat{\mathcal{T}} V_{t+1}(s,a)| \le \beta_t \cdot \big[ \phi(s_t, a_t)^\top \Lambda_t^{-1} \phi(s_t, a_t) \big]^{1/2}
\tag{38}
$$

for all $(s,a) \in \mathcal{S} \times \mathcal{A}$ with probability at least $1 - \xi$. $\qquad\square$

Theorem 2 allows us to further characterize the optimality gap of the pessimistic value iteration. In particular, the following corollary holds.

**Corollary 1** (Jin et al. (2021)). *Under the same conditions as Theorem 2, it holds that*

$$
V^*(s_1) - V^{\pi_1}(s_1) \le \sum_{t=1}^T \mathbb{E}_{\pi^*} \big[ \Gamma_t^{\mathrm{lcb}}(s_t, a_t) \,|\, s_1 \big]
$$

*Proof.* See e.g., Jin et al. (2021) for a detailed proof. $\qquad\square$

We remark that the optimality gap in Corollary 1 is information-theoretically optimal under the linear MDP setup with finite horizon (Jin et al., 2021). Intuitively, for an offline dataset with sufficiently good coverage on the optimal trajectories such as the experience from experts, such gap is small. For a dataset collected from random policy, such a gap can be large. Our experiments also support such intuition empirically, where the score obtained by training with the expert dataset is higher than that with the random dataset.

Theorem 2 shows that if we set $y = \mathcal{T} V_{t+1}(s^{\mathrm{ood}}, a^{\mathrm{ood}})$, then our estimation based on disagreement among ensembles is a valid $\xi$-uncertainty quantifier. However, such OOD target is impossible to obtain in practice as it requires knowing the transition at the OOD datapoint $(s^{\mathrm{ood}}, a^{\mathrm{ood}})$. In practice, if TD error is sufficiently minimized, then $Q_{t+1}(s, a)$ should well estimate the target $\mathcal{T} V_{t+1}$. Thus, in practice, we utilize

$$
y = Q_{t+1}(s^{\mathrm{ood}}, a^{\mathrm{ood}}) - \Gamma_{t+1}(s^{\mathrm{ood}}, a^{\mathrm{ood}})
$$

as the OOD target, where we introduce an additional penalization $\Gamma_{t+1}(s^{\mathrm{ood}}, a^{\mathrm{ood}})$ to enforce the pessimistic value estimation.

In addition, we remark that in theory, we require that the embeddings of the OOD sample are isotropic, in the sense that the eigenvalues of the corresponding covariate matrix $\Lambda_{\mathrm{ood}}$ are lower

bounded. Such isotropic property can be guaranteed by randomly generating states and actions. In practice, we find that randomly generating states is more expensive than randomly generating actions. Meanwhile, we observe that randomly generating actions alone are sufficient to guarantee reasonable empirical performance since the generated OOD embeddings are sufficiently isotropic. Thus, in our experiments, we randomly generate OOD actions according to our current policy and sample OOD states from the in-distribution dataset.

## B  IMPLEMENTATION DETAIL

### B.1  ALGORITHMIC DESCRIPTION

---
**Algorithm 1** PBRL algorithm
---
1: **Initialize:** $K$ bootstrapped $Q$-networks and target $Q$-networks with parameter $\theta$ and $\theta^-$, policy $\pi$ with parameter $\varphi$, and hyper-parameters $\beta_{\mathrm{in}}$, $\beta_{\mathrm{ood}}$
2: **Initialize:** total training steps $H$, current frame $h = 0$
3: **while** $h < H$ **do**
4:     Sample mini-batch transitions $(s, a, r, s')$ from the offline dataset $\mathcal{D}_{\mathrm{in}}$
5:     *# Critic Training for offline data.*
6:     Calculate the bootstrapped uncertainty $\mathcal{U}_{\theta^-}(s', a')$ through the target-networks.
7:     Calculate the $Q$-target in Eq. (4) and the resulting TD-loss $|Q_\theta^k(s, a) - \widehat{\mathcal{T}}^{\mathrm{in}} Q^k(s, a)|^2$.
8:     *# Critic Training for OOD data*
9:     Perform OOD sampling and obtains $N_{\mathrm{ood}}$ OOD actions $a^{\mathrm{ood}}$ for each $s$.
10:     Calculate the bootstrapped uncertainty $\mathcal{U}_\theta(s, a^{\mathrm{ood}})$ for OOD actions through $Q$-networks.
11:     Calculate the $Q$-target in Eq. (5) and the pseudo TD-loss $|Q_\theta^k(s, a^{\mathrm{ood}}) - \widehat{\mathcal{T}}^{\mathrm{ood}} Q^k(s, a^{\mathrm{ood}})|^2$.
12:     Minimize $|Q_\theta^k(s, a) - \widehat{\mathcal{T}}^{\mathrm{in}} Q^k(s, a)|^2 + |Q_\theta^k(s, a^{\mathrm{ood}}) - \widehat{\mathcal{T}}^{\mathrm{ood}} Q^k(s, a^{\mathrm{ood}})|^2$ to train $\theta$ by SGD.
13:     *# Actor Training*
14:     Improve $\pi_\varphi$ by maximizing $\min_k Q^k(s, a_\pi) - \log \pi_\varphi(a_\pi|s)$ with entropy regularization.
15:     Update the target $Q$-network via $\theta^- \leftarrow (1 - \tau)\theta^- + \tau\theta$.
16: **end while**
---

### B.2  HYPER-PARAMETERS

Most hyper-parameters of PBRL follow the SAC implementations in `https://github.com/rail-berkeley/rlkit`. We use the hyper-parameter settings in Table 2 for all the Gym domain tasks. We use different settings of $\beta_{\mathrm{in}}$ and $\beta_{\mathrm{ood}}$ for the experiment for Adroit domain and fix the other hyper-parameters the same as Table 2. See §E for the setup of Adroit. In addition, we use the same settings for discount factor, target network smoothing factor, learning rate, and optimizers as CQL (Kumar et al., 2020).

Table 2: Hyper-parameters of PBRL

| Hyper-parameters | Value | Description |
|---|---|---|
| $K$ | 10 | The number of bootstrapped networks. |
| $Q$-network | FC(256,256,256) | Fully Connected (FC) layers with ReLU activations. |
| $\beta_{\mathrm{in}}$ | 0.01 | The tuning parameter of in-distribution target $\widehat{\mathcal{T}}^{\mathrm{in}}$. |
| $\beta_{\mathrm{ood}}$ | $5.0 \rightarrow 0.2$ | The tuning parameter of OOD target $\widehat{\mathcal{T}}^{\mathrm{ood}}$. We perform linearly decay within the first 50K steps, and perform exponentially decay in the remaining steps. |
| $\tau$ | 0.005 | Target network smoothing coefficient. |
| $\gamma$ | 0.99 | Discount factor. |
| $lr$ of actor | 1e-4 | Policy learning rate. |
| $lr$ of critic | 3e-4 | Critic learning rate. |
| Optimizer | Adam | Optimizer. |
| $H$ | 1M | Total gradient steps. |
| $N_{\mathrm{ood}}$ | 10 | Number of OOD actions for each state. |

**Baselines.** We conduct experiments on D4RL with the latest 'v2' datasets. The dataset is released at `http://rail.eecs.berkeley.edu/datasets/offline_rl/gym_mujoco_v2_old/`. We now introduce the implementations of baselines. (i) The implementation of CQL (Kumar et al., 2020) is adopted from the official implementation at `https://github.com/aviralkumar2907/CQL`. In our experiment, we remove the BC warm-up stage since we find CQL performs better without warm-up for 'v2' dataset. (ii) For BEAR (Kumar et al., 2019), UWAC (Wu et al., 2021) and MOPO (Yu et al., 2020), we adopt their official implementations at `https://github.com/aviralkumar2907/BEAR`, `https://github.com/apple/ml-uwac`, and `https://github.com/tianheyu927/mopo`, respectively. We adopt their default hyper-parameters in training. (iii) Since the original paper of TD3-BC (Fujimoto & Gu, 2021) reports the performance of Gym in 'v2' dataset in the appendix section, we directly cite the reported scores in Table 1. The learning curve reported in Fig. 2 is trained by implementation released at `https://github.com/sfujim/TD3_BC`.

**Computational cost comparison.** In the sequel, we compare the computational cost of PBRL against CQL. We conduct such a comparison based on the Halfcheetah-medium-v2 task. We measure the number of parameters, GPU memory, and runtime per epoch (1K gradient steps) for both PBRL and CQL in the training. We run experiments on a single A100 GPU. We summarize the result in Table 3. We observe that, similar to the other ensemble-based methods such as Bootstrapped DQN (Osband et al., 2016), IDS (Nikolov et al., 2019), and Sunrize (Lee et al., 2021a), our method requires extra computation to handle the ensemble of $Q$-networks. In addition, we remark that a large proportion of computation for CQL is due to the costs of logsumexp over multiple sampled actions (Fujimoto & Gu, 2021), which we do not require.

Table 3: Comparison of computational costs.

|      | Runtime (s/epoch) | GPU memory | Number of parameters |
|------|-------------------|------------|----------------------|
| CQL  | 30.3              | 1.1G       | 0.42M                |
| PBRL | 52.1              | 1.7G       | 1.52M                |

### B.3 REMARKS ON FORMULATIONS OF PESSIMISM IN ACTOR AND CRITIC

We use different formulations to enforce pessimism in actor and critic. In the critic training, we use the penalized Q-function as in Eq. (4) and Eq. (5). While in the actor training, we use the minimum of ensemble Q-function as in Eq. (7). According to the analysis in EDAC [6], using the minimum of ensemble Q-function $\min_{j=1,\ldots,K} Q_j$ as the target is approximately equivalent to using $\bar{Q} - \beta_0 \cdot \mathrm{Std}(Q_j)$ with a fixed $\beta_0$ as the target. In contrast, in the critic training of PBRL, we tune different factors (i.e., $\beta_{\mathrm{in}}$ and $\beta_{\mathrm{ood}}$) for the in-distribution target and the OOD target, which yields better performance for the critic estimation.

In the actor training, since we already have pessimistic Q-functions learned by the critic, it is not necessary to enforce large penalties in the actor. To see such a fact, we refer to the ablation study in Fig. 9, where utilizing the mean as the target achieves reasonable performances. We remark that taking the minimum as the target avoids possible large values at certain state-action pairs, which may arise due to the numerical computation in fitting neural networks. As suggested by our ablation study, taking the minimum among the ensemble of Q-functions as the target achieves the best performance. Thus, we use the minimum among the ensemble of Q-functions as the target in PBRL.

## C   ABLATION STUDY

In this section, we present the ablation study of PBRL. In the training, we perform online interactions to evaluate the performance for every 1K gradient steps. Since we run each method for 1M gradient steps totally, each method is evaluated 1000 times in training. The experiments follow such evaluation criteria, and each curve is drawn by 1000 evaluated scores through online interaction.

**Number of bootstrapped-$Q$.**   We conduct experiments with different numbers of bootstrapped $Q$-networks in PBRL, and present the performance comparison in Fig. 4. We observe that the performance of PBRL is improved with the increase of the bootstrapped networks. We remark that since the training of offline RL is more challenging than online RL, it is better to use sufficient bootstrapped $Q$-functions to obtain a reasonable estimation of the non-parametric $Q$-posterior. We observe from Fig. 4 that using $K = 6, 8$, and $10$ yields similar final scores, and a larger $K$ leads to more stable performance in the training process. We adopt $K = 10$ for our implementation.

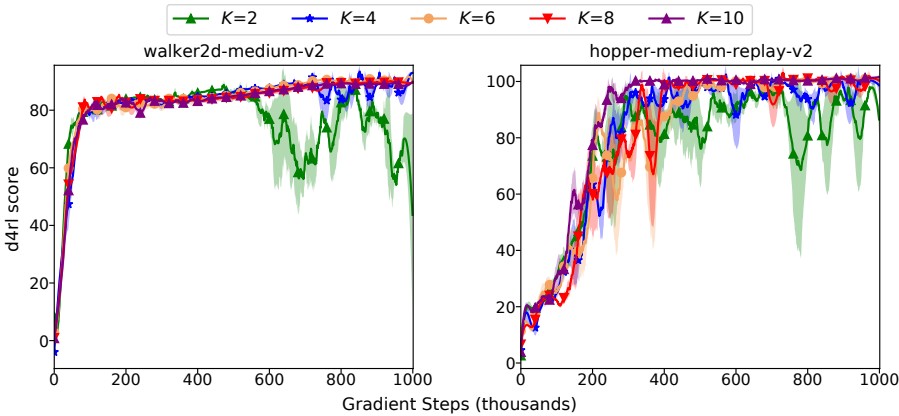

Figure 4: The Ablation on the number of bootstrapped $Q$-functions.

**Uncertainty of in-distribution target.**   We compare different kinds of uncertainty penalization in $\widehat{\mathcal{T}}^{\text{in}}$ for in-distribution data. (i) Penalizing the immediate reward only. (ii) Penalizing the next-$Q$ value only, which is adopted in PBRL. (iii) Penalizing both the immediate reward and next-$Q$ value. We present the comparison in Fig. 5. We observe that the target-$Q$ penalization performs the best, and adopt such penalization in the proposed PBRL algorithm.

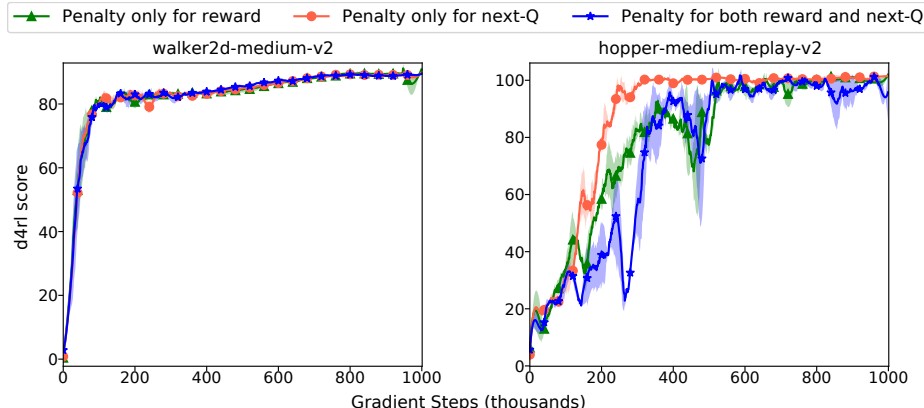

Figure 5: The Ablation on penalty strategy for the in-distribution data.

**Tuning parameter $\beta_{\text{in}}$.**   We conduct experiments with $\beta_{\text{in}} \in \{0.1, 0.01, 0.001, 0.0001\}$ to study the sensitivity of PBRL to the tuning parameter $\beta_{\text{in}}$ for the in-distribution target. We present the

comparison in Fig. 6. We observe that in the 'medium' dataset generated by a single policy, the performance of PBRL is insensitive to $\beta_{\text{in}}$. One possible reason is that since the 'medium' dataset is generated by a single policy, the offline dataset tends to concentrate around a few trajectories and has low uncertainty. Thus, the magnitude of $\beta_{\text{in}}$ has a limited effect on the penalty. Nevertheless, in the 'medium-replay' dataset, since the data is generated by various policies, the uncertainty of offline data is larger than that of the 'medium' dataset (as shown in Fig. 3). Correspondingly, the performance of PBRL is affected by the magnitude of $\beta_{\text{in}}$. Our experiment shows that PBRL performs the best for $\beta_{\text{in}} \in [0.0001, 0.01]$. We adopt $\beta_{\text{in}} = 0.01$ for our implementation.

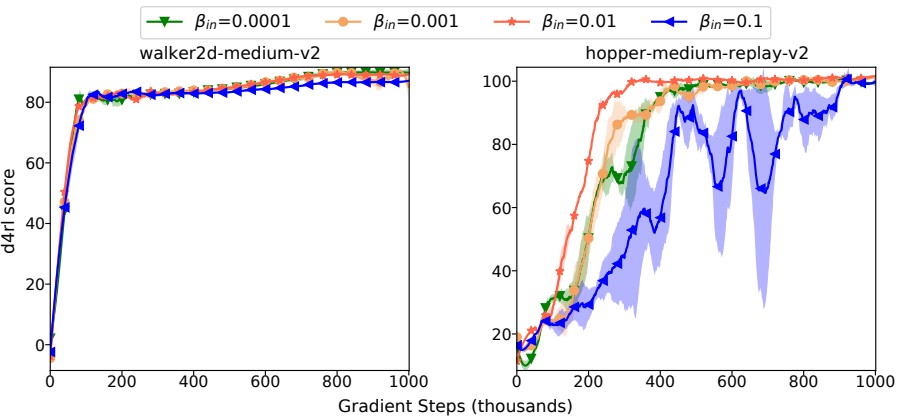

Figure 6: The Ablation on the tuning parameter $\beta^{\text{in}}$ in the in-distribution target.

**Tuning parameter $\beta_{\text{ood}}$.** We use a large $\beta_{\text{ood}}$ initially to enforce strong OOD regularization in the beginning of training, and then decrease $\beta_{\text{ood}}$ linearly while the training evolves. We conduct experiments with constant settings of $\beta_{\text{ood}} \in \{0.01, 0.1, 1.0\}$. We observe that in the 'medium' dataset, a large $\beta_{\text{ood}} = 1.0$ performs the best since the samples are generated by a fixed policy with a relatively concentrated in-distribution dataset. Thus, the OOD samples tend to have high uncertainty and are less trustworthy. In contrast, in the 'medium-replay' dataset, a small $\beta_{\text{ood}} \in \{0.1, 0.01\}$ performs reasonably well since the mixed dataset has larger coverage of the state-action space and the uncertainty of OOD data is smaller than that of the 'medium' dataset. Thus, adopting a smaller $\beta_{\text{ood}}$ for the 'medium-replay' dataset allows the agent to exploit the OOD actions and gain better performance. To match all possible situations, we propose the decaying strategy. Empirically, we find such a decaying strategy performs well in both the 'medium' and 'medium-replay' datasets.

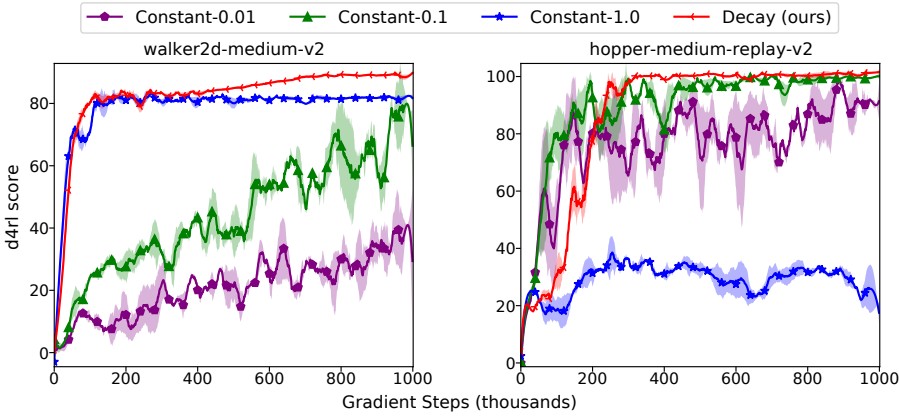

Figure 7: The Ablation on the tuning parameter $\beta^{\text{ood}}$ in the OOD target.

We also record the $Q$-value for $(s, a) \sim \mathcal{D}_{\text{in}}$ in the training process. As shown in Fig. 8, since both the in-distribution actions and OOD actions are represented by the same $Q$-network, a large con-

stant $\beta_{\text{ood}}$ makes the Q-value for in-distribution action overly pessimistic and leads to sub-optimal performance. It is desirable to use the decaying strategy for $\beta_{\text{ood}}$ in practice.

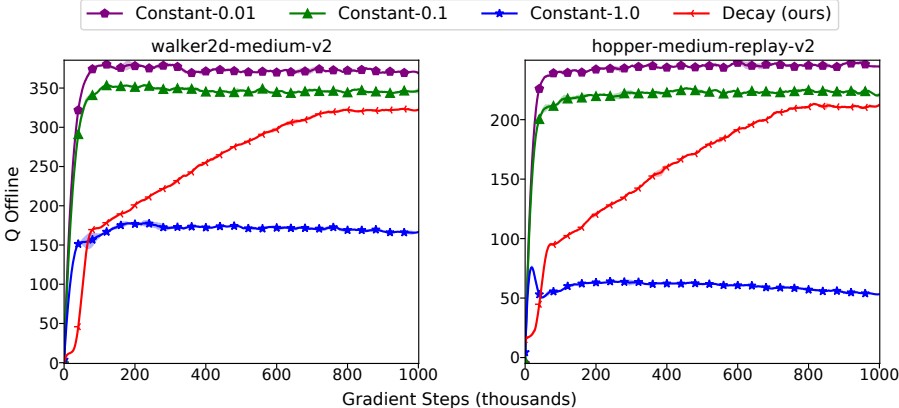

Figure 8: With different settings of $\beta^{\text{ood}}$, we show the Q-value for state-action pairs sampled from $\mathcal{D}_{\text{in}}$ in the training process.

**Actor training.** We evaluate different actor training schemes in Eq. (7), i.e., the actor follows the gradients of 'min' (in PBRL), 'mean' or 'max' value among $K$ Q-functions. The result in Fig. 9 shows training the actor by maximizing the 'min' among ensemble-$Q$ performs the best. In the 'medium-replay' dataset, since the uncertainty estimation is difficult in mixed data, taking 'mean' in the actor can be unstable in training. Taking 'max' in the actor performs worse in both tasks due to overestimation of $Q$-functions.

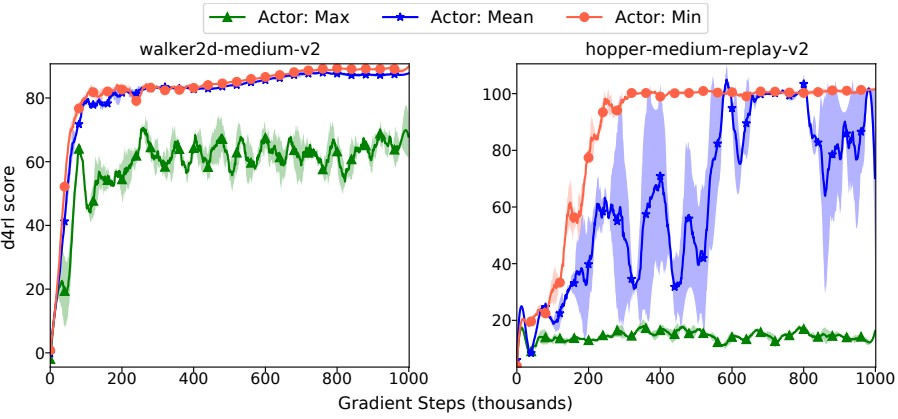

Figure 9: The Ablation on action selection scheme of the actor.

**Number of OOD samples.** The pessimism in critic training is implemented by sampling OOD actions and then performing uncertainty penalization, as shown in Eq. (6). In each training step, we perform OOD sampling and sample $N_{\text{ood}}$ actions from the learned policy. We perform an ablation study with $N_{\text{ood}} \in \{0, 2, 5, 10\}$. According to Fig 10, the performance is poor without OOD sampling (i.e., $N_{\text{ood}} = 0$). We find the performance becomes better with a small number of OOD actions (i.e., $N_{\text{ood}} = 2$). Also, PBRL is robust to different settings of $N_{\text{ood}}$.

**OOD Target.** In §A.3, we show that setting the learning target of OOD samples to zero enforces a Gaussian prior to $Q$-function under the linear MDP setting. However, in DRL, we find such a setup leads to overly pessimistic value function and performs poorly in practice, as shown in Fig. 11. Specifically, we observe that such a strong regularizer enforces the $Q$-functions to be close to zero for both in-distribution and OOD state-action pairs, as shown in Fig. 12 and Fig. 13. In contrast,

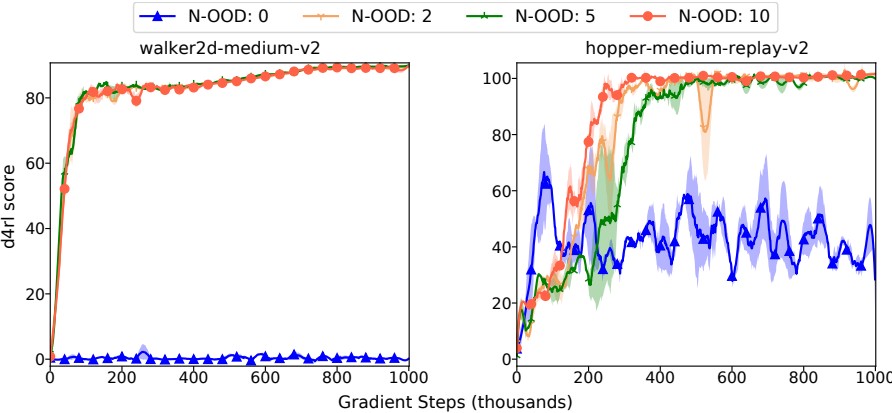

Figure 10: The Ablation on different number of OOD actions. the performance becomes better even with very small amout of OOD actions.

our proposed PBRL target performs well and does not yield large extrapolation errors, as shown in Fig. 12 and Fig. 13.

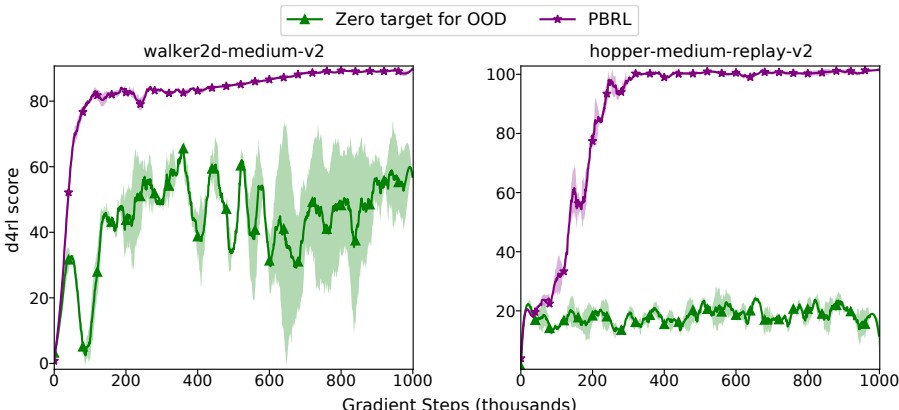

Figure 11: The Ablation on OOD target with $y^{\text{ood}} = 0$ (*normalized scores*).

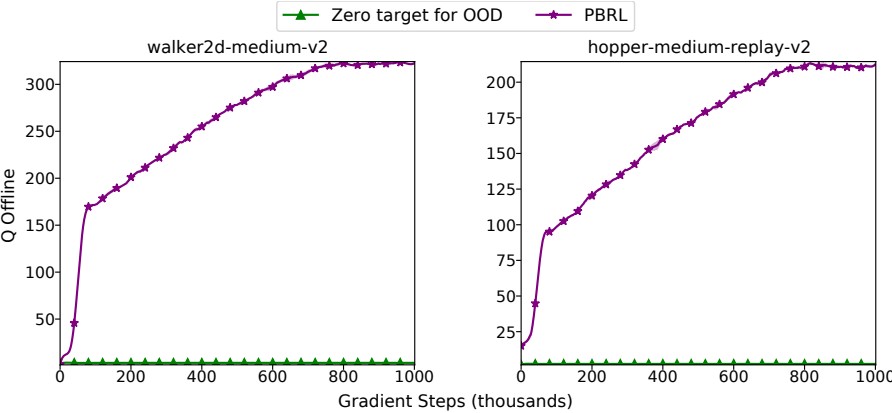

Figure 12: The Ablation on OOD target with $y^{\text{ood}} = 0$. $Q$-offline is the $Q$-value for $(s, a)$ pairs sampled from the offline dataset, where $a$ follows the behavior policy.

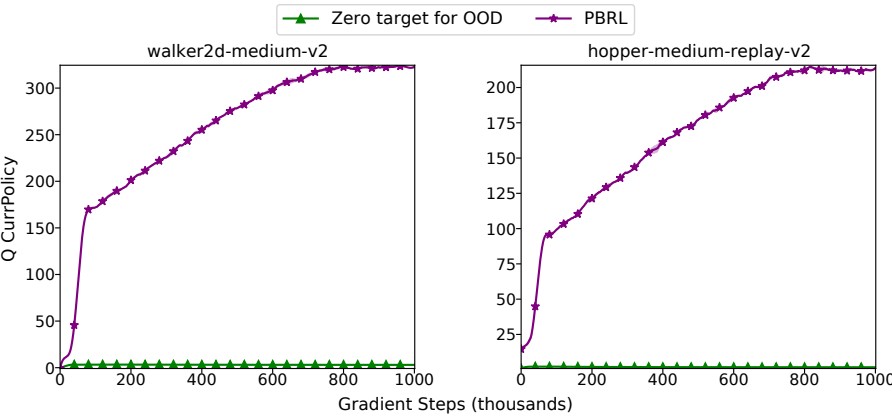

Figure 13: The Ablation on OOD target with $y^{\mathrm{ood}} = 0$. $Q$-CurrPolicy is the $Q$-value for $(s, a_\pi)$ pairs, where $a_\pi \sim \pi(a|s)$ follows the training policy $\pi$.

# D  REGULARIZATION FOR PBRL

In this section, we compare different regularization methods that are popular in other Deep Learning (DL) literature to solve the extrapolation error in offline RL. Specifically, we compare the following regularization methods.

- *None*. We remove the OOD-sampling regularizer in PBRL and train bootstrapped $Q$-functions solely based on the offline dataset through $\widehat{\mathcal{T}}^{\text{in}}$.

- *$L_2$-regularizer.* We remove the OOD-sampling regularizer and use $L_2$ normalization instead. As we discussed in §3.3, in linear MDPs, LSVI utilizes $L_2$ regularization to control the extrapolation behavior of $Q$-functions in OOD samples. In DRL, we add the $L_2$-norm of weights in the $Q$-network in loss functions to conduct $L_2$-regularization. We attempt two scale factors $\{1e-2, 1e-4\}$ in our experiments.

- *Spectral Normalization (SN).* We remove the OOD-sampling regularizer and use SN instead. SN constrains the Lipschitz constant of layers, which measures the smoothness of the neural network. Recent research (Gogianu et al., 2021) shows that SN helps RL training when applied to specific layers of the $Q$-network. In our experiment, we follow Gogianu et al. (2021) and consider two cases, namely, applying SN in the output layer (denoted by *SN[-1]*) and applying SN in both the output layer and the one before it (denoted by *SN[-1,-2]*), respectively.

- *Pessimistic Initialization (PI).* Optimistic initialization is simple and efficient for RL exploration, which initializes the $Q$-function for all actions with a high value. In online RL, such initialization encourages the agent to explore all actions in the interaction. Motivated by this method, we attempt a pessimistic initialization to regulate the OOD behavior of the $Q$-network. In our implementation, we draw the initial value of weights and a bias of the $Q$-networks from the uniform distribution $\text{Unif}(a, b)$. We try two settings in our experiment, namely, (i) *PI-small* that sets $(a, b)$ to $(-0.2, 0)$, and (ii) *PI-large* that sets $(a, b)$ to $(-1.0, 0)$. In both settings, we remove the OOD sampling and use PI instead.

We illustrate (i) the normalized performance in the training process, (ii) the $Q$-value along the trajectory of the training policy, and (iii) the uncertainty quantification along the trajectory of the training policy. We present the results in Fig. 14, Fig. 15, and Fig. 16, respectively. In the sequel, we discuss the empirical results.

- We observe that OOD sampling is the only regularization method with reasonable performance. Though $L_2$-regularization and SN yield reasonable performance in supervised learning, they do not perform well in offline RL.

- In the 'medium-replay' dataset, we observe that PI and SN can gain some score in the early stage of training. Nevertheless, the performance drops quickly along with the training process. We conjecture that both PI and SN have the potential to be effective with additional parameter tuning and algorithm design.

In conclusion, previous regularization methods for DL and RL are not sufficient to handle the distribution shift issue in offline RL. Combining such regularization techniques with policy constraints and conservatism methods may lead to improved empirical performance, which we leave for future research.

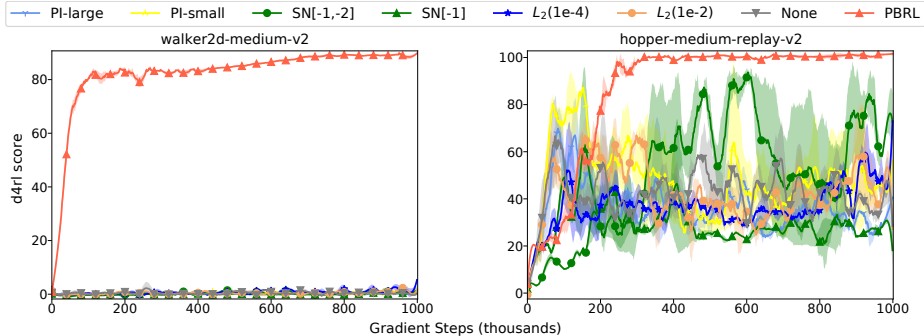

Figure 14: Comparision of different regularizers (*normalized score*)

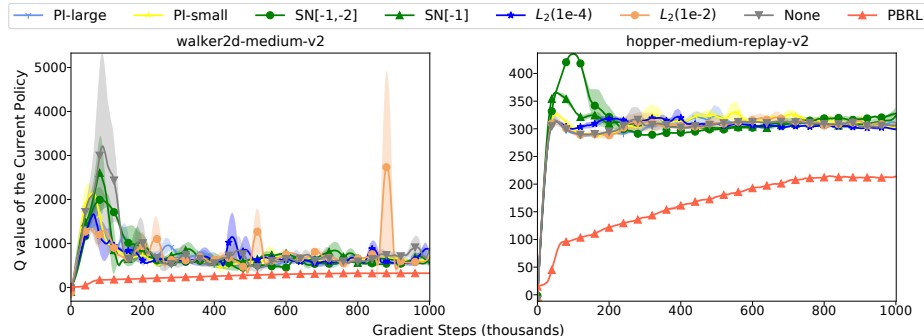

Figure 15: Comparision of different regularizers (*Q-value along trajectories of the training policy*)

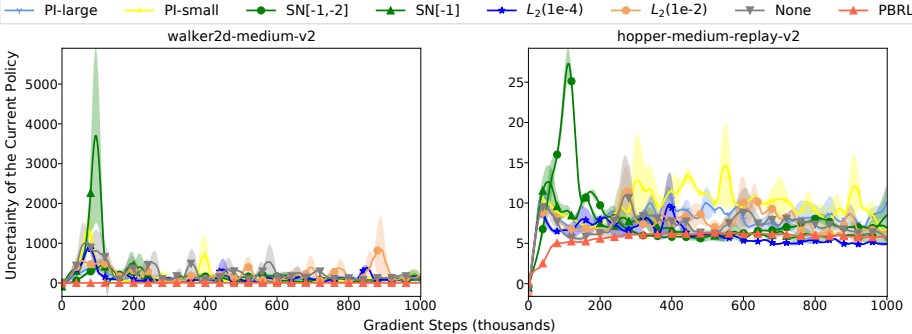

Figure 16: Comparision of different regularizers (*Uncertainty along trajectories of the training policy*)

# E    EXPERIMENTS IN ADROIT DOMAIN

In Adroit, the agent controls a 24-DoF robotic hand to hammer a nail, open a door, twirl a pen, and move a ball, as shown in Fig. 17. The Adroit domain includes three dataset types, namely, demonstration data from a human ('human'), expert data from an RL policy ('expert'), and fifty-fifty mixed data from human demonstrations and an imitation policy ('cloned'). The adroit tasks are more challenging than the Gym domain in task complexity. In addition, the use of human demonstration in the dataset also makes the task more challenging. We present the normalized scores in Table 4. We set $\beta_{in} = 0.0001$ and $\beta_{ood} = 0.01$. The other hyper-parameters in Adroit follows Table 2.

In Table 4, the scores of BC, BEAR, and CQL are adopted from the D4RL benchmark (Fu et al., 2020). We do not include the scores of MOPO (Yu et al., 2020) as it is not reported and we cannot find suitable hyper-parameters to make it work in the Adroit domain. We also attempt TD3-BC (Fujimoto et al., 2018) with different BC weights and fail in getting reasonable score for most of the tasks. In addition, since UWAC (Wu et al., 2021) has a different evaluation process, we re-train UWAC with the official implementation and report the scores in Table 4.

In Table 4, we add the average score without 'expert' dataset since the scores of 'expert' dataset dominate that of the rest of the datasets. We find CQL (Kumar et al., 2020) and BC have the best average performance among all baselines. Meanwhile, we observe that PBRL slightly outperforms CQL and BC. We remark that the human demonstrations stored in 'human' and 'cloned' are inherently different from machine-generated 'expert' data since (i) the human trajectories do not follow the Markov property, (ii) human decision may be affected by unobserved states such as prior knowledge, distractors, and the action history, and (iii) different people demonstrate differently as their solution policies. Such characteristic makes the 'human' and 'cloned' dataset challenging for offline RL algorithms. In addition, we remark that the recent study in robot learning from human demonstration also encounter such challenges (Mandlekar et al., 2021).

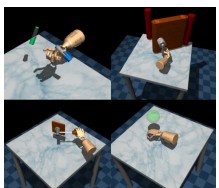

Figure 17: Illustration of tasks in Adroit domain.

|  |  | BC | BEAR | UWAC | CQL | MOPO | TD3-BC | **PBRL** |
|---|---|---|---|---|---|---|---|---|
| Human | Pen | 34.4 | -1.0 | 10.1 ±3.2 | 37.5 | - | 0.0 | 35.4 ±3.3 |
| | Hammer | 1.5 | 0.3 | 1.2 ±0.7 | 4.4 | - | 0.0 | 0.4 ±0.3 |
| | Door | 0.5 | -0.3 | 0.4 ±0.2 | 9.9 | - | 0.0 | 0.1 ±0.0 |
| | Relocate | 0.0 | -0.3 | 0.0 ±0.0 | 0.2 | - | 0.0 | 0.0 ±0.0 |
| Cloned | Pen | 56.9 | 26.5 | 23.0 ±6.9 | 39.2 | - | 0.0 | 74.9 ±9.8 |
| | Hammer | 0.8 | 0.3 | 0.4 ±0.0 | 2.1 | - | 0.0 | 0.8 ±0.5 |
| | Door | -0.1 | -0.1 | 0.0 ±0.0 | 0.4 | - | 0.0 | 4.6 ±4.8 |
| | Relocate | -0.1 | -0.3 | -0.3 ±0.0 | -0.1 | - | 0.0 | -0.1 ±0.0 |
| Expert | Pen | 85.1 | 105.9 | 98.2 ±9.1 | 107.0 | - | 0.3 | 137.7 ±3.4 |
| | Hammer | 125.6 | 127.3 | 107.7 ±21.7 | 86.7 | - | 0.0 | 127.5 ±0.2 |
| | Door | 34.9 | 103.4 | 104.7 ±0.4 | 101.5 | - | 0.0 | 95.7 ±12.2 |
| | Relocate | 101.3 | 98.6 | 105.5 ±3.2 | 95.0 | - | 0.0 | 84.5 ±12.2 |
| Average | | 36.73 | 38.41 | 37.57 ±3.8 | 40.31 | - | 0.02 | 46.79 ±3.9 |
| Average w/o expert | | 11.74 | 3.21 | 4.35 ±1.4 | 11.70 | - | 0.02 | 14.52 ±2.3 |

Table 4: Average normalized score over 3 seeds in Adroit domain. The highest performing scores are highlighted. Among all methods, PBRL and CQL outperform the best of the baselines.

## F    RELIABLE EVALUATION FOR STATISTICAL UNCERTAINTY

Recent research (Agarwal et al., 2021) proposes reliable evaluation principles to address the statistical uncertainty in RL. Since the ordinary aggregate measures like *mean* can be easily dominated by a few outlier scores, Agarwal et al. (2021) presents several efficient and robust alternatives that are not unduly affected by outliers and have small uncertainty even with a handful of runs. In this paper, we adopt these evaluation methods for each method in Gym domain with $M_{\text{task}} * N_{\text{seed}}$ runs.

- *Stratified Bootstrap Confidence Intervals.* The Confidence Intervals (CIs) for a finite-sample score estimates the plausible values for the true score. Bootstrap CIs with stratified sampling can be applied to small sample sizes and is better justified than sample standard deviations.

- *Performance Profiles.* Performance profiles reveal performance variability through score distributions. A score distribution shows the fraction of runs above a certain score and is given by $\hat{F}(\tau) = \hat{F}(\tau; x_{1:M,1:N}) = \frac{1}{M} \sum_{m=1}^{M} \frac{1}{N} \sum_{n=1}^{N} \mathbb{1}[x_{m,n} \geq \tau]$.

- *Aggregate Metrics.* Based on bootstrap CIs, we can extract aggregate metrics from score distributions, including median, mean, interquartile mean (IQM), and optimality gap. IQM discards the bottom and top 25% of the runs and calculates the mean score of the remaining 50% runs. Optimality gap calculates the amount of runs that fail to meet a minimum score of $\eta = 50.0$.

The result comparisons are give in Fig. 18 and Fig. 19. Specifically, Fig. 18 shows aggregate metrics based on 95% bootstrap CIs, and Fig. 19 shows performance profiles based on score distribution. For both evaluations, our PBRL and PBRL-prior outperforms other methods with small variability.

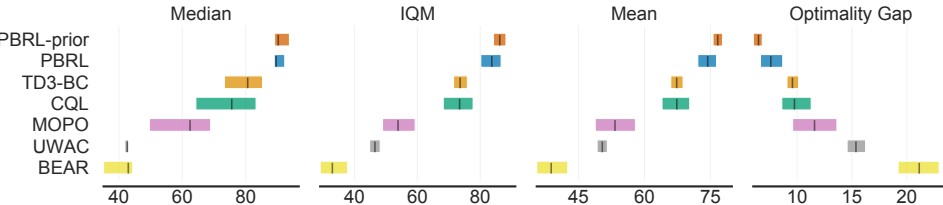

Figure 18: Aggregate metrics on D4RL with 95% CIs based on 15 tasks and 5 seeds for each task. Higher mean, median and IQM scores, and lower optimality gap are better. The CIs are estimated using the percentile bootstrap with stratified sampling. Our methods perform better than baselines.

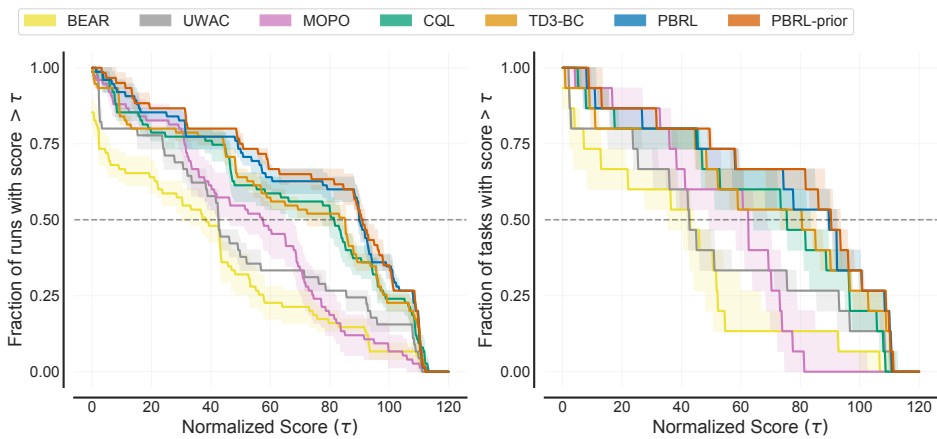

Figure 19: Performance profiles on D4RL based on score distributions (left), and average score distributions (right). Shaded regions show pointwise 95% confidence bands based on percentile bootstrap with stratified sampling. The $\tau$ value where the profiles intersect $y = 0.5$ shows the median, and the area under the performance profile corresponds to the mean.

