# OpenReview forum: "Pessimistic Bootstrapping for Uncertainty-Driven Offline Reinforcement Learning"
_ICLR.cc/2022/Conference — ICLR 2022 Spotlight_

### Official Review · Reviewer_73UJ · 2021-10-29

**Correctness:** 3
**Technical Novelty And Significance:** 2
**Empirical Novelty And Significance:** 2
**Recommendation:** 6
**Confidence:** 4

**Main Review:**

Strengths:

-- This specific algorithm is novel, although borrows some elements from existing works (which are all referenced properly, as far as I can tell).

-- Experiments are setup up properly and the algorithm shows nice results compared to baselines.

-- The writing is clear and easy to follow.

Weaknesses:

-- I feel that some of the theoretical results in the main text are misleading, especially when compared to what is actually proved in the appendix. Namely, Lemma 1 in the main text asserts that the bootstrapped uncertainties in the algorithm are equivalent to LCB penalties at the heart of recent theoretical works. In contrast, the appendix establishes a completely different equivalence, showing that when performing Bayesian linear regression, the posterior is related to the same LCB penalties used in recent works. There is no established equivalence between Bayesian linear regression and the bootstrapped uncertainties used in the proposed algorithm, and in fact, I don't think there would be any equivalence, since in the only variation/diversity introduced into the critic ensemble is via initialization and SGD, which are irrelevant in the linear closed-form setting.

-- The OOD term of the critic loss seems strange to me. It is effectively regressing each ensemble member Q_i to Q_i - uncertainty(Q), so the Q_i's will be encouraged to decrease indefinitely, until uncertainty(Q) becomes zero somehow. In fact, in a tabular setting, the gradient update would decrease each Q_i by the same amount, learning_rate * uncertainty(Q), and so the uncertainty will be unchanged over SGD updates, and so the Q_i's will diverge to negative infty.

-- I would be interested to see more ablations (I may have missed some of these); e.g., the same algorithm but without the OOD term in the critic loss, or perhaps replacing the OOD penalty with some other penalty (CQL-style penalty?). Perhaps also measuring the uncertainty in other ways -- previous works (BEAR, BCQ) backup a minimum over an ensemble; is this better than penalizing with stddev?

**Summary Of The Paper:**

The paper looks at the offline RL problem and claims that existing techniques penalize out-of-distribution actions too aggressively and so perform poorly at generalization and extrapolation. To remedy this, the paper proposes an offline RL algorithm in which an ensemble of critics is trained with an objective composed of (1) a TD error based on actions seen in the dataset with target value penalized by standard deviation over the ensemble of the next Q-values, and (2) a squared error on the Q-values of OOD actions regressing to those same Q-values penalized by the standard deviation over the ensemble. The proposed algorithm is paired with a theoretical analysis connecting it to recent theoretical offline RL papers. The algorithm is evaluated on the D4RL benchmarks and shows favorable performance compared to baselines.

**Summary Of The Review:**

Overall, the paper is a nice contribution, but I think some aspects of the presentation need to be changed before it is ready for publication.

---

> ### Author Response · Authors · 2021-11-22
> **Response to Reviewer 73UJ**
>
>
> We thank the reviewer for the valuable comments and time dedicated to evaluating our work.
>
> **Comment 1:** The connection between Bayesian linear regression (BLR) and bootstrapped uncertainties.
>
> **Response 1:** Lemma 1 is an informal statement that captures the Lemma 2 in Appendix A.2, which is rigorously proved. In our revision, we change Lemma 1 into a claim and add a clarification regarding the claim. We remark that Lemma 1 establishes the connections between the frequentist confidence interval and the posterior standard deviation. Meanwhile, various existing works connect the bootstrapped ensemble and the Bayesian posterior with uninformative prior [1,2,3].
>
> In addition, the randomized ensemble without bootstrapping is a practical and efficient implementation that estimates the bootstrapped ensemble [1,2,7,8]. Specifically, in [1,2], it is observed that the randomized ensemble without bootstrapping can achieve a similar performance as the bootstrapped ensemble. In addition, recent work also justifies such an empirical finding from the theoretical perspective. See, e.g., [3,4,5,6] for the analysis.
>
> ----
>
> **Comment 2:** The OOD term of the critic loss seems strange to me... $Q_i$'s will diverge to negative infty.
>
> **Response 2:** As we discussed in Theorem 1, for an OOD state-action pair $(s^{\rm ood}, a^{\rm ood})$, if we set the regression target as $\mathcal{T}V(s^{\rm ood}, a^{\rm ood})$, the bootstrapped uncertainty evaluated on the disagreement (e.g., standard deviation) within the ensemble is a valid $\xi$-uncertainty quantifier. Upon incorporating an uncertainty penalty, the regression target becomes
>
> $\mathcal{T}V(s^{\rm ood}, a^{\rm ood})-\beta_{\rm ood}\mathcal{U}(s^{\rm ood},a^{\rm ood})=\big[r(s^{\rm ood}, a^{\rm ood})-\beta_{\rm ood}\mathcal{U}(s^{\rm ood},a^{\rm ood})\big]+\gamma\mathbb{E}_P[V (s')].$
>
> Intuitively, one can consider the uncertainty penalty, $-\beta_{\rm ood}\mathcal{U}(s^{\rm ood},a^{\rm ood})$,  as a negative term incorporated into the current reward $r(s^{\rm ood},a^{\rm ood})$ — it does not affect the contraction property of the penalized Bellman operator, which arises from the discounted total future reward $\gamma\mathbb{E}_P[V (s')]$, especially the contraction parameter $\gamma\in [0,1)$ therein. Meanwhile, since the transition kernel for the next state $s'$ given the current state-action pair $(s^{\rm ood}, a^{\rm ood})$ is unknown a priori, we use the current Q-value $Q(s^{\rm ood}, a^{\rm ood})$ to approximate $\mathcal{T}V(s^{\rm ood}, a^{\rm ood})$. Such an approximation is sufficiently accurate if the TD-error is sufficiently small, which in practice is ensured by the convergence of the policy evaluation algorithm for learning the critic. In other words, given that the TD-error is sufficiently small, the penalized Bellman operator $\widehat{\mathcal{T}}^{\rm ood}$ still has the contraction property.
>
> In conclusion, if the TD-error is sufficiently minimized, the operator $\widehat{\mathcal{T}}^{\rm ood}$ still has a  contraction property. Nevertheless, in the training process, such a TD-error is not guaranteed to be sufficiently small. In our implementation, we introduce several additional approaches to ensure that the $Q$-function does not diverge, which are summarized as follows.
>
> - The OOD action $a^{\rm ood}$ is sampled from the current policy $\pi$, which is updated in each training step. Thus, the uncertainty penalty is applied based on actions with a constantly changing distribution rather than a fixed distribution.
>
> - As we discussed in Section 3.2 (last paragraph on Page 4) and Figure 7 in Appendix C, we reduce the factor $\beta_{\rm ood}$ for $\widehat{\mathcal{T}}^{\rm ood}Q^{\rm ood}$ penalty gradually in the training process to weaken the penalization for OOD actions gradually. We add an additional visualization in Figure 8 in Appendix C for such penalization. We remark that if we use a large constant $\beta_{\rm ood}$ for $\widehat{\mathcal{T}}^{\rm ood}Q^{\rm ood}$, the Q-value for the offline data becomes overly pessimistic and leads to sub-optimal performances. Hence, it is important to use a decaying strategy for $\beta_{\rm ood}$.
>
> - In addition, our implementation has an extra truncation term to ensure that the OOD target is always positive. Specifically, in our implementation, the OOD target takes the form of $\max(\widehat{\mathcal{T}}^{\rm ood}Q^{\rm ood},0)$. We refer to our released code for the details. We remark that such truncation is inactive in training, as suggested in Figure 3 of the manuscript. Thus, such truncation does not lead to bias in fitting the Q-functions.

---

> > ### Author Response · Authors · 2021-11-22
> > **Response to Reviewer 73UJ (follow up)**
> >
> > **Comment 3:** I would be interested to see more ablations.
> >
> > **Response 3:** Thanks for the question about the ablations.
> >
> > *[The same algorithm but without the OOD term in the critic loss?]* This ablation has been included in Appendix D and is denoted as "None".
> >
> > *[Replacing the OOD penalty with some other penalty (CQL-style penalty?)]* We remark that our ensemble of critic is proposed to estimate the uncertainty quantifier. Nevertheless, such uncertainty quantifier is unnecessary if one adopt CQL-style critic. Hence, we conjecture that the ensemble-type CQL would not bring us extra insight to compare with.
> >
> > *[previous works (BEAR, BCQ) backup a minimum over an ensemble; is this better than penalizing with stddev?]* We remark that although BEAR uses minimum over an ensemble, BEAR requires minimizing the MMD distance between behavior policy and the current policy to regularize the behavior of critics. Empirically, BEAR cannot work without such a behavior constraint.  In addition, BCQ also requires such a behavior constraint to regularize the behavior of critics. In contrast, PBRL is a purely uncertainty-based method without such policy constraints.
> >
> > In addition, we remark that using the minimum of ensemble Q-function $\min_{j=1,\dots,K} Q_j$ as the target is approximately equivalent to using $\bar{Q}-\beta_0 \cdot {\rm Std}(Q_j)$ with a fixed $\beta_0$ as the target [9]. In contrast, in the critic training of PBRL, we tune different factors (i.e., $\beta_{\rm in}$ and $\beta_{\rm ood}$) for the in-distribution target and the OOD target, which yields better performance for the critic estimation.
> >
> > In the actor training, since we already have pessimistic Q-functions learned by the critic, it is not necessary to enforce large penalties in the actor. To see such a fact, we refer to the ablation study in Fig. 9, where utilizing the mean as the target achieves reasonable performances. We remark that taking the minimum as the target avoids possible large values at certain state-action pairs, which may arise due to the numerical computation in fitting neural networks. As suggested by our ablation study in Fig. 9, taking the minimum among the ensemble of Q-functions as the target achieves the best performance. Thus, we use the minimum among the ensemble of Q-functions as the target in PBRL. We refer to Appendix B.3 in our revision for the details.
> >
> > ------
> >
> > References
> >
> > [1] Ian Osband, Charles Blundell, Alexander Pritzel, and Benjamin Van Roy. Deep Exploration via Bootstrapped DQN. In NeurIPS 2016.
> >
> > [2] Ian Osband, John Aslanides, and Albin Cassirer. Randomized Prior Functions for Deep Reinforcement Learning. In NeurIPS 2018.
> >
> > [3] Balaji Lakshminarayanan, Alexander Pritzel, and Charles Blundell. Simple and Scalable Predictive Uncertainty Estimation Using Deep Ensembles. In NeurIPS 2017.
> >
> > [4] Yiding Jiang, Vaishnavh Nagarajan, Christina Baek, and J. Zico Kolter. Assessing Generalization of SGD via Disagreement. arXiv preprint. 2021.
> >
> > [5] Botao Hao, Xiang Ji, Yaqi Duan, Hao Lu, Csaba Szepesvári, and Mengdi Wang. Bootstrapping Statistical Inference for Off-Policy Evaluation. arXiv preprint. 2021.
> >
> > [6] Botao Hao, Yasin Abbasi-Yadkori, Zheng Wen, and Guang Cheng. Bootstrapping Upper Confidence Bound. In NeurIPS 2019.
> >
> > [7] Nikolay Nikolov, Johannes Kirschner, Felix Berkenkamp, and Andreas Krause. Information-Directed Exploration for Deep Reinforcement Learning. In ICLR 2019.
> >
> > [8] Kimin Lee, Michael Laskin, Aravind Srinivas, and Pieter Abbeel. Sunrise: A Simple Unified Framework for Ensemble Learning in Deep Reinforcement Learning. In ICML 2021.
> >
> > [9] Gaon An, Seungyong Moon, Jang-Hyun Kim, and Hyun Oh Song. Uncertainty-Based Offline Reinforcement Learning with Diversified Q-Ensemble. In  NeurIPS 2021.

---

> > > ### Comment · Reviewer_73UJ · 2021-11-24
> > > **Thanks**
> > >
> > > Thanks for your response. I am glad that your "Claim 1" is now more appropriately presented.

---

### Official Review · Reviewer_wgNz · 2021-10-31

**Correctness:** 4
**Technical Novelty And Significance:** 1
**Empirical Novelty And Significance:** 3
**Recommendation:** 6
**Confidence:** 4

**Main Review:**

The paper offers theoretical support in the linear MDP setting and has compelling empirical results in the D4RL suite (with both MuJoCo and Adroit tasks). Comments:

1. Can the authors spell out all assumptions for the theoretical section of the paper when describing the problem setting?

2. The paper presents why the proposed bonuses can be viewed as a lower confidence bound type penalty. The paper doesn't present a result comparing the value of the learnt policy against the optimal policy under more assumptions (e.g. Bellman completeness and an appropriate coverage/relative condition number assumption). See Chapter 3 in Agarwal et al. and the work of Zanette et al. as a means to obtain such guarantees.

3. The assumption on \Lambda_ood having eigenvalues lowerbounded by lambda appears rather strong (and rather extraneous). Comments on why this is needed or how it can be relaxed?

4. The theory and practical sections don't particularly relate to each other in that the theory focuses on linear/tabular MDPs. One way to bridge this divide would be through working out uncertainty measures under a Kernel Nonlinear Regulator (KNR) assumption, see [Mania et al., Kakade et al.].

5. Regarding hyper-parameters:
a. Are the hyper-parameters in table 2 used in all environments considered in the paper?
b. What were the hyper-parameters that exhibited the most sensitivity? Could you mention what ranges these values these hyper-parameters were grid searched on?
c. For a given hyper-parameter, did you have to perform online interactions after every few gradient steps to examine the value of the policy that you have, or did you perform online interactions once per hyper-parameter (at the end of learning)?
Could you detail answers to these questions in the appendix?

4. Could you mention what Q offline and Q current policy mean in figure 10?

Agarwal et al. Reinforcement Learning: Theory and Algorithms
Zanette et al. Provable Benefits of Actor-Critic Methods for offline RL
Kakade et al. Information Theoretic Regret Bounds for Online Nonlinear Control
Mania et al. Certainty equivalent control of LQR is efficient.


**Summary Of The Paper:**

This paper presents a model-free pessimistic bootstrapping approach for offline RL. Specifically, the paper considers an actor-critic approach with an ensemble of Q-functions and utilizing disagreements in their predictions (measured as standard deviation) for learning the Q-functions. The paper also presents a way to regularize the learning with out of distribution state-action pairs which according to the paper is a crucial part in obtaining improved results.

---
Update post author response: I have increased my score.

**Summary Of The Review:**

The theory results in their current form are limited, and this can be addressed by expanding its scope to include classes of non-linear dynamics models; furthermore, the theory results are partial in their characterization since they fall short of characterizing the sub-optimality of the learnt policy as a function of properties of the offline dataset (measured in the form of coverage). That being said, the paper is definitely promising in terms of its empirical results. I'd be up for revisiting my reviews if these issues can be addressed.

---

> ### Author Response · Authors · 2021-11-22
> **Response to Reviewer wgNz**
>
> We thank the reviewer for the valuable comments and time dedicated to evaluating our work.
>
> **Comment 1:** Can the authors spell out all assumptions for the theoretical section of the paper when describing the problem setting?
>
> **Response 1**: In Theorem 1, we impose the linear MDP assumption to justify that OOD sampling leads to a valid $\xi$-uncertainty quantifier. We remark that the linear MDP assumption is commonly adopted in the theoretical analysis of RL with function approximation [1,8]. Moreover, from a technical perspective,  the linear setting can be readily extended to the (infinite-dimensional) kernelized linear setting or the neural network setting via the neural tangent kernel regime [5]. The analysis therein only replaces the dimension in the linear setting with the effective dimension. In addition, for (infinite-dimensional) kernel and neural network settings, we can establish similar arguments showing that OOD sampling leads to a valid uncertainty quantifier.
>
> The linear MDP assumption is already sufficient if we have a closed-form $\xi$-uncertainty quantifier. In practice, however, we do not have a closed-form $\xi$-uncertainty quantifier in the neural network setting (which is not necessarily in the neural tangent kernel regime). To this end, we derive a practical implementation in a principled manner (Claim 1) by exploiting the connection among the frequentist confidence region, the posterior standard deviation in Bayesian regression (e.g., under the Gaussian prior in the linear setting), which is proportional to the width of the frequentist confidence region, and the disagreement within the ensemble.
> The connection between the frequentist confidence region and the posterior standard deviationis well known in linear regression [13]. By exploiting such a connection, we argue the equivalence between the standard deviation within the ensemble, which is commonly used to approximate the posterior [9,10,11], and the valid $\xi$-uncertainty quantifier in the linear setting. In such settings, we can establish similar arguments showing that OOD sampling leads to a valid $\xi$-uncertainty quantifier.
>
> ------
>
> **Comment 2**: The paper presents why the proposed bonuses can be viewed as a lower confidence bound type penalty. The paper doesn't present a result comparing the value of the learned policy against the optimal policy under more assumptions (e.g., Bellman completeness and an appropriate coverage/relative condition number assumption). See Chapter 3 in Agarwal et al. and the work of Zanette et al. as a means to obtain such guarantees.
>
> **Response 2**: We thank the reviewer for the recent references on the theory of offline RL, most of which appeared during the writing of this paper.
> We have included a summary of the recent advances in the theory of offline RL on page 7. We have an optimality gap analysis under the linear MDP assumption in Corollary 1, page 18 of the paper. We remark that our analysis hinges on the linear MDP assumption, which is only slightly stronger than the Bellman completeness assumption. The reason is that linear MDP assumption implies Bellman completeness with respect to the linearly parameterized function class for all policies.
>
> Meanwhile, we remark that under an additional regularity condition on the feature map, the Bellman completeness with respect to the linear class for all policies implies that the underlying MDP is a linear MDP (as shown in, e.g., Proposition 5.1 in [1]). In other words, the two assumptions are almost equivalent except for an additional regularity condition.
>
> We remark that implementing the global notion of pessimism (and optimisim) on the Q-value of the initial state under the Bellman completeness assumption requires solving nested optimization problems [3, 4, 12], which remains challenging to implement in practice. In comparison, under the slightly stronger linear MDP assumption, we are able to utilize the notion of local pessimism (and optimisim), that is, the lower (and upper) confidence bound on the Q-value of all the state-action pairs, which is computationally tractable to implement by connecting the frequentist confidence region, posterior, and ensemble as mentioned above.
>
> In addition, we remark that our assumption imposed on the offline dataset is similar to the coverage/relative condition number assumptions proposed under the Bellman completeness assumption [2,3,4]. we refer to [this latex link](https://www.dropbox.com/s/oba1yk62tf4mep1/PBRL_rebuttal_proof.pdf?dl=0) for the details.

---

> > ### Author Response · Authors · 2021-11-22
> > **Response to Reviewer wgNz (follow-up)**
> >
> > **Comment 3:** The assumption on $\Lambda_{\rm ood}$ having eigenvalues lower bounded by lambda appears rather strong (and rather extraneous). Comments on why this is needed or how it can be relaxed?
> >
> > **Response 3:** We remark that such an assumption is imposed only on OOD samples, which consists of only state-action pairs and does not require a policy to sample. Under the linear MDP setting with normalized state-action embeddings, such a condition holds if one sample OOD embeddings by following a uniform distribution over the state-action embeddings. In addition, we remark that we introduce OOD sampling to emulate the $\ell_2$-regularization under the linear setting. We remark that although $\ell_2$-regularization is equivalent with the OOD sampling under the linear setting, our empirical study suggests that OOD sampling performs much better than the naive $\ell_2$-regularization under the neural network parameterization, where controlling the extrapolation error is more challenging.
> >
> > ------
> >
> > **Comment 4:** The theory and practical sections don't particularly relate to each other in that the theory focuses on linear/tabular MDPs. One way to bridge this divide would be through working out uncertainty measures under a Kernel Nonlinear Regulator (KNR) assumption, see [Mania et al., Kakade et al.].
> >
> > **Response 4:** We remark that the linear/tabular setting is introduced to simplify the discussion. It is possible to generalize the analysis to a more general kernelized linear setting or the neural network setting in the neural tangent kernel regime (e.g.,[5]) by replacing the dimension of the feature with the effective dimension. We thank the reviewer for pointing out the KNR model (e.g.,[6, 7]). KNR models the transition dynamics as Gaussian distributions with a linearly parameterized mean function class. By the definition of KNR, solving RL under the KNR setting requires model-based RL and an optimal planning oracle. We remark that it is unclear how to solve KNR via the model-free approach. In addition, solving KNR with model-based RL requires optimal planning, which is not computationally efficient. In contrast, our proposed method is model-free, which does not require such an optimal planning oracle.
> >
> > We agree that the KNR model is highly relevant and have added a discussion on the KNR model in Section 4 of our revision.
> >
> > ------
> >
> > **Comment 5:** Regarding hyper-parameters.
> >
> > **Response 5:** (a) The hyper-parameters given in Table 2 are used for all Gym tasks. We use slightly different hyper-parameters for the Adroit tasks. We remark that we adopt the same discount factor, target network smoothing factor, learning rate, and optimizers as CQL. We refer to Appendix E for the details.
> >
> > (b) We have an ablation study of hyper-parameters in Appendix C. We refer to Appendix C for the grid search range of hyper-parameters. We find that the most sensitive hyper-parameters is $\beta_{\rm ood}$, which controls the uncertainty penalty for OOD actions. As we discussed in Section 3.2 and Appendix C of our manuscript, the uncertainty quantifiers are inaccurate in the early stage of training. Thus, we propose to use a large $\beta_{\rm ood}$ in the beginning and then gradually decrease it during the training process. We find that such a decaying strategy of $\beta_{\rm ood}$ performs well among all tasks.
> >
> > (c) In the training process, we perform online interactions to evaluate the performance for every 1K of gradient steps. Since we train each method for 1M gradient steps in total, each method is evaluated 1000 times in the training. All the figures in Appendix C are plotted based on such online evaluations. We added more details on the evaluation in Appendix C of our revision.
> >
> > ------
> >
> > **Comment 6:** Could you mention what Q offline and Q current policy mean in Figure 10?
> >
> > **Response 6:** We thank the reviewer for pointing out the unclear notation in Figure 10. Q-offline is the Q-value for $(s,a)$ pairs sampled from the offline dataset, where $a$ follows the behavior policy. Q-CurrPolicy is the Q-value for $(s,a_{\pi})$ pairs, where $a_{\pi}\sim \pi(a|s)$ follows the training policy $\pi$. In our revision, we added an explanation for the notation in the caption.

---

> > > ### Author Response · Authors · 2021-11-22
> > > **References**
> > >
> > > References
> > >
> > > [1] Chi Jin, Zhuoran Yang, Zhaoran Wang, and Michael I. Jordan. Provably Efficient Reinforcement Learning with Linear Function Approximation. In COLT. 2020
> > >
> > > [2] Masatoshi Uehara, Xuezhou Zhang, and Wen Sun. Representation Learning for Online and Offline RL in Low-rank MDPs. arXiv preprint arXiv:2110.04652. 2021.
> > >
> > > [3] Tengyang Xie, Ching-An Cheng, Nan Jiang, Paul Mineiro, and Alekh Agarwal. Bellman-consistent Pessimism for Offline Reinforcement Learning. In NeurIPS 2021.
> > >
> > > [4] Andrea Zanette, Martin J. Wainwright, and Emma Brunskill. Provable Benefits of Actor-Critic Methods for Offline Reinforcement Learning. In NeurIPS 2021.
> > >
> > > [5] Zhuoran Yang, Chi Jin, Zhaoran Wang, Mengdi Wang, and Michael I. Jordan. On Function Approximation in Reinforcement Learning: Optimism in the Face of Large State Spaces. In NeurIPS 2020.
> > >
> > > [6] Sham Kakade, Akshay Krishnamurthy, Kendall Lowrey, Motoya Ohnishi, and Wen Sun. Information Theoretic Regret Bounds for Online Nonlinear Control. In NeurIPS 2020.
> > >
> > > [7] Horia Mania, Michael I. Jordan, and Benjamin Recht. Active Learning for Nonlinear System Identification with Guarantees. arXiv preprint  arXiv:2006.10277. 2020.
> > >
> > > [8] Ying Jin, Zhuoran Yang, and Zhaoran Wang. Is Pessimism Provably Efficient for Offline RL? In ICML 2021.
> > >
> > > [9] Ian Osband, Charles Blundell, Alexander Pritzel, and Benjamin Van Roy. Deep Exploration via Bootstrapped DQN. In NeurIPS 2016.
> > >
> > > [10] Ian Osband, John Aslanides, and Albin Cassirer. Randomized Prior Functions for Deep Reinforcement Learning. In NeurIPS 2018.
> > >
> > > [11] Balaji Lakshminarayanan, Alexander Pritzel, and Charles Blundell. Simple and Scalable Predictive Uncertainty Estimation Using Deep Ensembles. In NeurIPS 2017.
> > >
> > > [12] Aditya Modi, Jinglin Chen, Akshay Krishnamurthy, Nan Jiang, and Alekh Agarwal. Model-Free Representation Learning and Exploration in Low-Rank
> > > MDPs. arXiv preprint arXiv:2102.07035. 2021.
> > >
> > > [13]  Mike West. Outlier models and prior distributions in Bayesian linear regression. Journal of the Royal Statistical Society: Series B (Methodological). 1984.

---

> > > > ### Comment · Reviewer_wgNz · 2021-11-22
> > > > **Re: Author Response**
> > > >
> > > > Thanks to the authors for their detailed notes.  Based on their response, I have increased my score.
> > > >
> > > > 1. At a high level, I am still somewhat bothered by lowerbounded eigenvalue assumption on the ood (s,a) pairs - i think in general, these issues can be mitigated by utilizing pessimism. Without this, it seems like there is some assumption of full coverage of ood pairs on all of the state-action space, which isn't really going to be satisfied in the offline RL setting (see remarks 4/7/11 in [1] and references therein).
> > > >
> > > > 2. With regards to the sub-optimality bound - I see corollary 1 in page 18, but, relating this to some form of relative condition number bound is what i was expecting - such a bound helps obtain a finite sample rate that contains the relative condition number, number of offline samples, horizon, dimension of the features. For instance, see a similar result in [1] but in the context of imitation learning (and references therein).
> > > >
> > > > If the authors can commit to addressing these issues (perhaps in the final revision), or saying why such a result cannot be obtained, I'd be open to improving my scores even further.
> > > >
> > > > [1] Chang et al. Mitigating Covariate Shift in Imitation Learning via Offline Data Without Great Coverage (2021).

---

> > > > > ### Author Response · Authors · 2021-11-23
> > > > > **Follow-up Response to Reviewer wgNz**
> > > > >
> > > > > **Response 1:** We remark that in theory, we require that the embeddings of the OOD sample are isotropic, in the sense that the eigenvalues of the corresponding covariate matrix $\Lambda_{\mathrm{ood}}$ are lower bounded. Such isotropic property can be guaranteed by randomly generating states and actions. In practice, we find that randomly generating states is more expensive than randomly generating actions. Meanwhile, we observe that randomly generating actions alone are sufficient to guarantee reasonable empirical performances since the generated OOD embeddings are sufficiently isotropic. Thus, in our experiments, we randomly generate OOD actions according to our current policy and sample OOD states from the in-distribution dataset.
> > > > >
> > > > > ------------
> > > > >
> > > > > **Response 2:** We can extend our optimality gap guarantee under the sufficient coverage assumption in [1]. Specifically, under the assumption that $\tilde \Lambda \succeq I + c\cdot n \cdot \Lambda^*(s)$ with high probability for all $s\in S$, where $\Lambda^*(s) = \mathbb{E}_{\pi^*}[\phi(s, a)\phi(s, a)^\top | s_0 = s]$ and $\tilde \Lambda$ is the covariate matrix defined in (9) of the manuscript, the optimality gap scales as $O(1/\sqrt{n})$ with high probability. We refer to Corollary 4.5 of [1] for the details.
> > > > >
> > > > > [1] Ying Jin, Zhuoran Yang, and Zhaoran Wang. Is Pessimism Provably Efficient for Offline RL? in ICML 2021

---

> > > > > > ### Comment · Reviewer_wgNz · 2021-11-30
> > > > > > **Response to author comment**
> > > > > >
> > > > > > right, my understanding is that the O(1/sqrt(n)) rate should have factors analogous to the concentrability coefficient in the numerator. I think adding in guarantee in the main paper would be great. thanks!

---

> > > > > > > ### Author Response · Authors · 2021-12-01
> > > > > > > **Response**
> > > > > > >
> > > > > > > Thank you!
> > > > > > >
> > > > > > > Indeed there is a concentrability coefficient. Following Corollary 4.5 in [1] (and our previous response), suppose $\tilde \Lambda_n \succeq I + n \cdot C^* \Lambda ^* $, then it holds that the suboptimality gap is bounded by $$C\cdot d^{1.5} H^2 \sqrt{C^*} \cdot n^{-0.5} \cdot \sqrt{ \log (4dHn / \zeta)}$$ with probability at least $1 - \zeta$.
> > > > > > > Here $\zeta \ in (0,1)$, $d$ is the dimension of the feature mapping, $H$ is the horizon length, $C$ is an absolute constant. And more importantly, $C*$ is the _concentrability coefficient_ which measures how well the _offline dataset_ covers the _trajectory of the optimal policy_.

---

### Official Review · Reviewer_hc93 · 2021-11-02

**Correctness:** 4
**Technical Novelty And Significance:** 2
**Empirical Novelty And Significance:** 3
**Recommendation:** 8
**Confidence:** 4

**Main Review:**

**Pros**
- This paper is well-written and easy to read.
- The paper provides the theoretic interpretation of their uncertainty penalization.
- The proposed method outperforms various offline RL algorithms including model-based algorithms and uncertainty-aware methods.
- The paper provides informative ablation studies comparing various regularizations and analyzing the effects of hyperparameters.

**Cons**
- The proposed method requires additional computation costs and memory for Q-network ensemble and OOD penalization. It will be informative to compare methods in terms of computational cost and space.
- Formulation of actor-critic objectives is relying on heuristics. For example, the critic loss term uses uncertainty penalized Q-values, whereas the policy loss term uses the minimum Q-value. Is there any reason to use these different forms?

**Additional comments**
- Does the PBRL use identical hyperparameter settings provided in Table 2 across all of the Gym and Adroit domains? If then, it is very surprising, because each task has a different dimension and complexity. As I experienced, most of the existing offline RL algorithms including CQL, MOPO, etc. are very sensitive to hyperparameters (e.g, lr, penalty coefficient) and must tune their hyperparameters to each domain. Because it is hard to validate models without online interaction, I believe the robustness to hyperparameters is a very important issue in Offline RL.
- There is a recent paper [1] that is highly related to this submission. The proposed method seems more stable, but however, it will be informative to compare the proposed method with [1] in revision.

[1] An et. al, Uncertainty-Based Offline Reinforcement Learning with Diversified Q-Ensemble, To appear at NeurIPS 2021 (https://arxiv.org/abs/2110.01548).

**Post Rebuttal**
I appreciate the author's efforts and clarification. The paper becomes much clearer (e.g., hyperparameter settings) and improved by adding computation cost analysis and other baselines. Thus I maintain my score.



**Summary Of The Paper:**

This paper proposed an uncertainty-aware offline reinforcement learning algorithm based on a Q-network ensemble. The proposed method penalizes Q-values on OOD actions and performs pessimistic offline Q-learning. The paper interprets the method with LCB framework. The proposed method outperforms existing offline RL algorithms on D4RL Gym and Adroit tasks.

**Summary Of The Review:**

This paper is well motivated and clearly written with sufficient theoretical and experimental backups. Overall, I believe this is a good paper.

---

> ### Author Response · Authors · 2021-11-22
> **Response to Reviewer hc93**
>
> We thank the reviewer for the valuable comments and time dedicated to evaluating our work.
>
> **Comment 1:** Additional computation costs and memory.
>
> **Response 1:** We added a comparison between PBRL and CQL in training the Halfcheetah-medium-v2 task. In our comparison, we measure the number of parameters, GPU memory, and runtime per epoch (1K gradient steps) for the experiments. We run experiments on a single A100 GPU. We summarize the result in the following table. We observe that similar to the other ensemble-based methods, such as Bootstrapped DQN [2,3], IDS [4], and Sunrize [5], our method requires extra computation to handle the ensemble of Q-networks. In addition, we remark that a large proportion of computation for CQL is due to the costs of computing the log-sum-exp function over multiple sampled actions [1], which we do not require. We have added our comparison in Appendix B of our revision.
>
> |         | Runtime (s/epoch) | GPU memory | number of parameters |
> |  ----   | ---- | ---- | ---- |
> | CQL     | 30.3 | 1.1G | 0.42M |
> | PBRL    | 52.1 | 1.7G | 1.52M |
>
> ------
>
> **Comment 2:** The critic loss term uses uncertainty penalized Q-values, whereas the policy loss term uses the minimum Q-value. Is there any reason to use these different forms?
>
> **Response 2:** EDAC [6] proposes an uncertainty-based offline RL method with the ensemble-diversified critic. According to Eq.(3) in EDAC, if the ensemble of Q-functions are Gaussian, using the minimum of ensemble Q-function $\min_{j=1,\dots,K} Q_j$ as the target is approximately equivalent to using $\bar{Q}-\beta_0 \cdot {\rm Std}(Q_j)$ with a fixed $\beta_0$ as the target. In contrast, in the critic training of PBRL, we tune different factors (i.e., $\beta_{\rm in}$ and $\beta_{\rm ood}$) for the in-distribution target and the OOD target, which yields better performance for the critic estimation.
>
> In the actor training, since we already have pessimistic Q-functions learned by the critic, it is not necessary to enforce large penalties in the actor. To see such a fact, we refer to the ablation study in Fig. 9, where utilizing the **mean** as the target achieves reasonable performances. We remark that taking the minimum as the target avoids possible large values at certain state-action pairs, which may arise due to the numerical computation in fitting neural networks. As suggested by our ablation study in Fig. 9, taking the minimum among the ensemble of Q-functions as the target achieves the best performance. Thus, we use the minimum among the ensemble of Q-functions as the target in PBRL. We refer to Appendix B.3 in our revision for the details.
>
> ------
>
> **Comment 3:** Does the PBRL use identical hyper-parameter settings provided in Table 2 across all of the Gym and Adroit domains?
>
> **Response 3:** In our experiments, we use the **same** hyper-parameter settings provided in Table 2 for all Gym domain tasks. The experiment for the Adroit domain has different $\beta_{\rm in}$ and $\beta_{\rm ood}$ settings, whereas the other hyper-parameters are identical to that in Table 2. See Appendix E for the experimental setup of the Adroit domain.
>
> As a comparison, MOPO needs different hyper-parameters (e.g., rollout length, penalty scale) for different tasks. Meanwhile, for CQL, we use the same hyper-parameter settings for all tasks, similar to the setup of our proposed PBRL. In addition, in our experiment, we use a modified CQL by removing the *warm-up* (behavior cloning) stage in the official CQL implementation. We find CQL without such *warm-up* performs more robust to hyper-parameters and performs well in D4RL Gym v2 tasks with an identical hyper-parameter setting. In addition, our experiments on TD3-BC also use identical hyper-parameter setting for all Gym tasks. We refer the implementation details to Appendix B of our revision.
>
> We also refer to Appendix C for an ablation study of hyper-parameters. We observe that among all hyper-parameters, $\beta_{\rm ood}$ is the most sensitive parameter. To handle such a challenge, we propose a decaying strategy for $\beta_{\rm ood}$. As shown in Figure 7 in our manuscript, the decaying strategy performs well among all tasks.
>
> A possible explanation for the robustness of PBRL to hyper-parameters is that PBRL uses an ensemble critic with $10$ networks, which makes PBRL more robust to hyper-parameters compared with methods with fewer Q-networks. As shown in Figure 4 in Appendix C and in EDAC [6], more bootstrapped Q-functions result in a more stable performance empirically.

---

> > ### Author Response · Authors · 2021-11-22
> > **Response to Reviewer hc93 (follow up)**
> >
> >
> > **Comment 4:** There is a recent paper [6] that is highly related to this submission.
> >
> > **Response 4:** We thank the reviewer for the reference. We remark that EDAC [6] is a concurrent work and is released after the ICLR submission. Thus, we are unable to compare our work with EDAC in the initial version.
> >
> > Similar to our proposed PBRL, EDAC is an uncertainty-based method. Both EDAC and PBRL use the ensemble of Q-networks to ensure pessimism. The difference between the two algorithms is the approach for handling OOD actions. EDAC calculates the gradients of each Q-network and then diversifies such gradients to obtain sufficient penalization for OOD actions. In contrast, PBRL penalizes the OOD actions based on OOD sampling and the learned uncertainty quantifiers. Empirically, EDAC needs more Q-networks (10-50, see Figure 5 in [6]) than PBRL (6-10, see Figure 4 in our manuscript). We have added more discussions in Section 4 of our revision.
> >
> > ------
> >
> > References
> >
> > [1] Scott Fujimoto, and Shixiang Shane Gu. A Minimalist Approach to Offline Reinforcement Learning. In NeurIPS 2021
> >
> > [2] Ian Osband, Charles Blundell, Alexander Pritzel, and Benjamin Van Roy. Deep Exploration via Bootstrapped DQN. In NeurIPS 2016
> >
> > [3] Ian Osband, John Aslanides, and Albin Cassirer. Randomized Prior Functions for Deep Reinforcement Learning. In NeurIPS 2018
> >
> > [4] Nikolay Nikolov, Johannes Kirschner, Felix Berkenkamp, and Andreas Krause. Information-Directed Exploration for Deep Reinforcement Learning. In ICLR 2019
> >
> > [5] Kimin Lee, Michael Laskin, Aravind Srinivas, and Pieter Abbeel. Sunrise: A Simple Unified Framework for Ensemble Learning in Deep Reinforcement Learning. In ICML 2021.
> >
> > [6] Gaon An, Seungyong Moon, Jang-Hyun Kim, and Hyun Oh Song. Uncertainty-Based Offline Reinforcement Learning with Diversified Q-Ensemble. In  NeurIPS 2021.

---

### Official Review · Reviewer_HkNt · 2021-11-03

**Correctness:** 4
**Technical Novelty And Significance:** 3
**Empirical Novelty And Significance:** 4
**Recommendation:** 8
**Confidence:** 4

**Main Review:**

The paper is well written and easy to follow. The proposed PBRL is well motivated and is also backed by theoretical results in a linear MDP setting. The pessimistic value backup itself may not be new in the offline RL context (e.g. Appendix E in [1]), but it was introduced as a practical trick in the previous work, while PBRL's pessimistic bootstrapping is supported by theory, which I think is a good contribution. Also, to my knowledge, the way of exploiting the OOD samples is novel. Experiments and ablation studies are convincing and thorough. Overall, I think the paper made a solid contribution.

- In contrast to $\hat T^{in}$ in Eq (4), $\hat T^{ood}$ in Eq (5) is not a contraction mapping. Therefore, repeatedly applying $\hat T^{ood}$ would yield a divergence to $-\infty$ of Q-value, which seems problematic?


[1] Lee et al., Batch Reinforcement Learning with Hyperparameter Gradients, ICML 2020



**Summary Of The Paper:**

This paper presents Pessimistic Bootstrapping for offline RL (PBRL), a model-free offline RL algorithm that purely relies on an uncertainty-driven method. Bootstrapped Q-functions are trained, and the standard deviation of their estimates is used for the uncertainty quantification. This uncertainty quantification is then used for pessimistic bootstrapping. Also, in contrast to the existing methods that only considers in-distribution target, PBRL optimizes Q-function even for out-of-distribution actions with the pseudo-target that is penalized by the uncertainty quantifier. A theoretical analysis is provided that PBRL is provably efficient in the linear MDP setting. Experimental results demonstrate that PBRL outperforms the state-of-the-art methods.


**Summary Of The Review:**

The paper is well written. The proposed method is well-backed by theory and the empirical results are convincing.

---

> ### Author Response · Authors · 2021-11-22
> **Response to Reviewer HkNt**
>
> We thank the reviewer for the valuable comments and time dedicated to evaluating our work.
>
> **Comment 1**: $\widehat{\mathcal{T}}^{\rm ood}$ is not a contraction mapping.
>
> **Response 1**: As we discussed in Theorem 1, for an OOD state-action pair $(s^{\rm ood}, a^{\rm ood})$, if we set the regression target as $\mathcal{T}V(s^{\rm ood}, a^{\rm ood})$, the bootstrapped uncertainty evaluated on the disagreement (e.g., standard deviation) within the ensemble is a valid $\xi$-uncertainty quantifier. Upon incorporating an uncertainty penalty, the regression target becomes
>
> $\mathcal{T}V(s^{\rm ood}, a^{\rm ood})-\beta_{\rm ood}\mathcal{U}(s^{\rm ood},a^{\rm ood})=\big[r(s^{\rm ood}, a^{\rm ood})-\beta_{\rm ood}\mathcal{U}(s^{\rm ood},a^{\rm ood})\big]+\gamma\mathbb{E}_P[V (s')].$
>
> Intuitively, one can consider the uncertainty penalty, $-\beta_{\rm ood}\mathcal{U}(s^{\rm ood},a^{\rm ood})$,  as a negative term incorporated into the current reward $r(s^{\rm ood},a^{\rm ood})$ — it does not affect the contraction property of the penalized Bellman operator, which arises from the discounted total future reward $\gamma\mathbb{E}_P[V (s')]$, especially the contraction parameter $\gamma\in [0,1)$ therein. Meanwhile, since the transition kernel for the next state $s'$ given the current state-action pair $(s^{\rm ood}, a^{\rm ood})$ is unknown a priori, we use the current Q-value $Q(s^{\rm ood}, a^{\rm ood})$ to approximate $\mathcal{T}V(s^{\rm ood}, a^{\rm ood})$. Such an approximation is sufficiently accurate if the TD-error is sufficiently small, which in practice is ensured by the convergence of the policy evaluation algorithm for learning the critic. In other words, given that the TD-error is sufficiently small, the penalized Bellman operator $\widehat{\mathcal{T}}^{\rm ood}$ still has the contraction property.
>
> -----
>
> **Comment 2**: Repeatedly applying $\widehat{\mathcal{T}}^{\rm ood}$ would yield a divergence to $-\infty$ of Q-value, which seems problematic?
>
>
> **Response 2**: As suggested in Response 1, if the TD-error is sufficiently minimized, the operator $\widehat{\mathcal{T}}^{\rm ood}$ still has a  contraction property. Nevertheless, in the training process, such a TD-error is not guaranteed to be sufficiently small. In our implementation, we introduce several additional approaches to ensure that the Q-function does not diverge, which are summarized as follows.
>
> - The OOD action $a^{\rm ood}$ is sampled from the current policy $\pi$, which is updated in each training step. Thus, the uncertainty penalty is applied based on actions with a constantly changing distribution rather than a fixed distribution.
>
> - As we discussed in Section 3.2 (last paragraph on Page 4) and Figure 7 in Appendix C, we reduce the factor $\beta_{\rm ood}$ for $\widehat{\mathcal{T}}^{\rm ood}Q^{\rm ood}$ penalty gradually in the training process to weaken the penalization for OOD actions gradually. We add an additional visualization in Figure 8 in Appendix C for such penalization. We remark that if we use a large constant $\beta_{\rm ood}$ for $\widehat{\mathcal{T}}^{\rm ood}Q^{\rm ood}$, the Q-value for the offline data becomes overly pessimistic and leads to sub-optimal performances. Hence, it is important to use a decaying strategy for $\beta_{\rm ood}$.
>
> - In addition, our implementation has an extra truncation step to ensure that the OOD target is always nonnegative. Specifically, in our implementation, the OOD target takes the form of $\max(\widehat{\mathcal{T}}^{\rm ood}Q^{\rm ood},0)$. We refer to our released code for the details. We remark that the truncation is inactive in training, since the underlying true Q-function is always nonnegative. In addition, as suggested in Figure 3 of the manuscript, the fitted Q-function is also nonnegative. Thus, such truncation does not lead to bias in fitting the Q-functions.
>
> ------
>
> **Comment 3**: Pessimistic value backup in related work [1].
>
> **Response 3**: We thank the reviewer for pointing out the related work on BOPAH [1]. We remark that BOPAH is a model-based approach, whereas our method is model-free. In addition, although BOPAH uses uncertainty penalization, BOPAH requires an additional policy constraint to ensure that the learned policy does not deviate too far away from the behavior policy. In contrast, our method is purely uncertainty-driven without policy constraints. We remark that our method is provably efficient under linear MDPs.
>
> In addition, although we did not compare PBRL with BOPAH directly, we compare PBRL with TD3-BC, which so far achieves the best performance within the class of offline RL algorithms with policy constraints. We have added discussions on BOPAH in Section 4 of our revision.
>
> ----
>
> [1] Byungjun Lee, Jongmin Lee, Peter Vrancx, Dongho Kim, and Kee-Eung Kim. Batch Reinforcement Learning with Hyperparameter Gradients. In ICML 2020.
>
> [2] Scott Fujimoto, and Shixiang Shane Gu. A Minimalist Approach to Offline Reinforcement Learning. In NeurIPS 2021

---

> > ### Comment · Reviewer_HkNt · 2021-11-30
> > **Reply**
> >
> > Thanks for your response. I think the current notation of Eq (5) in its form is a bit confusing.
> > It looks like the operator $\hat T^{ood}: \mathbb{R}^{S \times A} \rightarrow \mathbb{R}^{S \times A}$ works as shifting a function $Q: \mathbb{R}^{S \times A}$ by $- \beta_{ood} \mathcal{U}_\theta$.
> >
> > This means:
> >
> > $( \hat{T}^{ood} \hat{T}^{ood} Q)(s,a) = Q(s,a) - 2 \beta_{ood} \mathcal{U}\theta(s,a)$,
> >
> > $( \hat{T}^{ood} \hat{T}^{ood} \hat{T}^{ood} Q)(s,a) = Q(s,a) - 3 \beta_{ood} \mathcal{U}_\theta(s,a)$
> > , and so on.
> >
> > This shifting operator would not be a contraction mapping.
> >
> > Furthermore, the terms like $\mathcal{T}V(s^{\rm ood}, a^{\rm ood})-\beta_{\rm ood}\mathcal{U}(s^{\rm ood},a^{\rm ood})=\big[r(s^{\rm ood}, a^{\rm ood})-\beta_{\rm ood}\mathcal{U}(s^{\rm ood},a^{\rm ood})\big]+\gamma\mathbb{E}_P[V (s')]$ do not appear until Eq (5) in the main text.
> > To prevent misunderstanding, it would be great to include more clarifications regarding contraction property when defining $\hat{T}^{ood}$ in Eq (5).

---

> > > ### Author Response · Authors · 2021-11-30
> > > **Follow-up response**
> > >
> > > Thanks for the reply. Our clarification in the previous response considers the Bellman target $\mathcal{T}^{\rm ood}$ as follows,
> > >
> > > $\mathcal{T}^{\rm ood}Q(s, a)=\big[r(s, a)-\beta_{\rm ood}\mathcal{U}(s,a)\big]+\gamma\mathbb{E}_P[V (s')].$
> > >
> > > If the TD error is sufficiently small, such target can be estimated by $Q_{\textrm{target}}(s, a) -\beta_{\rm ood}\mathcal{U}(s,a)$. In addition, we remark that in our implementation, the factor $\beta_{\rm ood}$ in OOD target has a decaying factor $\alpha<1$, which ensures that iteratively applying $\mathcal{T}^{\rm ood}$ does not diverge. Specifically, after applying $\mathcal{T}^{\rm ood}$ for $n$-times, the $Q$-function becomes ${(\mathcal{T}^{\rm ood})^{n}}Q(s,a)=Q(s,a)-\sum_{i=0}^{n-1} \alpha^{i} \mathcal{U}(s,a)$. As a result, when $n$ goes to infinity, we have ${(\mathcal{T}^{\rm ood})^{n}}Q(s,a)=Q(s,a)-\mathcal{U}(s,a)/(1-\alpha)$, which does not diverge to negative infinity. We refer to Section 3.2 and Appendix C for the details on the parameter $\beta_{\rm ood}$.
> > >
> > > In addition, we remark that our OOD target introduces the term $\sum_{(s, a)\in D_{\rm ood}}(Q(s, a) - Q(s, a) + \beta_{\rm ood}\mathcal{U}(s, a))^2 = \sum_{(s, a)\in D_{\rm ood}} \beta_{\rm ood}^2 \mathcal{U}^2(s, a)$ in the loss function. Such an extra term can be seen as a regularizer, which aims to minimize the uncertainty on the OOD state-action pairs. We will clarify the setup of the parameter $\beta_{\rm ood}$ and the motivation of OOD target in our revision.

---

> > > > ### Comment · Reviewer_HkNt · 2021-11-30
> > > > **Thanks**
> > > >
> > > > Thanks for your further clarification, which addressed my initial concern.
> > > >
> > > > I have a follow-up question regarding minimizing $\sum_{(s, a)\in D_{\rm ood}}(Q(s, a) - Q(s, a) + \beta_{\rm ood}\mathcal{U}(s, a))^2 = \sum_{(s, a)\in D_{\rm ood}} \beta_{\rm ood}^2 \mathcal{U}^2(s, a)$.
> > > > It seems that minimizing $ \sum_{(s, a)\in D_{\rm ood}} \beta_{\rm ood}^2 \mathcal{U}_{\theta}^2(s, a)$ does not always yield a penalty to $Q$ for OOD $(s,a)$. To reduce the standard deviation of the Q ensemble values, some values of $Q^k$ might *increase* towards $\bar Q$. Is my understanding correct? If so, why is it helpful to make each Q ensemble value have a similar value (rather than penalizing uncertain state-action)?

---

> > > > > ### Author Response · Authors · 2021-11-30
> > > > > **Follow-up response**
> > > > >
> > > > > Thanks for your reply.
> > > > >
> > > > > (1) $\mathcal{U}$ is the standard deviation of the ensemble Q-network.  In regression, $\mathcal{U}$ is fixed and considered as the response in the regression problem. The target of regression is $Q^{\rm target} - \mathcal{U}$, which is a pessimistic version of the regression target. So by using this operator, we introduce pessimism on OOD state-actions.
> > > > >
> > > > > (2) In our problem, the least-squares loss in fact should be $(Q^{\rm target} - \mathcal{U} - Q)^2$ (We simplified the notation in the previous response). That means, suppose we let $Q=Q^{\rm target}$, then the loss will be $\mathcal{U}^2$. To minimize the loss, we need to shrink the value of $Q$. Such a shrinkage is the "penalization" we mentioned in the previous response. The effect of this penalty is on the Q network instead of $Q^{\rm target}$.

---

> > > > > > ### Comment · Reviewer_HkNt · 2021-12-01
> > > > > > **Thanks**
> > > > > >
> > > > > > Thank you for your clarification.
> > > > > > To prevent potential confusion, it would be good to use an explicit notation for the target network in Eq 5.
> > > > > >
> > > > > > In Eq 4, the RHS is using $\theta^-$ explicitly, but in Eq 5, the RHS is using $\theta$ (rather than $\theta^-$).

---

> > > > > > > ### Author Response · Authors · 2021-12-01
> > > > > > > **Thanks**
> > > > > > >
> > > > > > > Thanks for your reply. We will modify our notation and add clarifications for the notation in our revision.

---

### Public Comment · ~Rishabh_Agarwal2 · 2021-11-10
**Suggestion for reliable evaluation**

Hi authors,

It seems that the average score across D4RL reported in the paper is easily prone to performance on outlier tasks -- for example, performing well on  a few tasks can lead to high average score.  Furthermore, It seems that a lot of the performance comparisons have overlaps in their standard deviation.

A way to more reliably report performances might be to show the performance profile with CIs of scores all the tasks and seeds (which uses the scores on 15 tasks x 5 seeds/task = 75 seeds) to show the distribution of scores across different tasks (mean/mean/percentiles can be read from such plots). Similarly, aggregate metrics which are less prone to outliers than mean (such as interquartile mean / median / optimality gap) with bootstrap confidence intervals that depend on the entire 75 seeds can be reported to improve the statistical confidence in the results. Since you have access to scores on individual seeds, you can easily incorporate the above recommendations using the library at https://github.com/google-research/rliable or the corresponding [colab](https://bit.ly/statistical_precipice_colab).

[1] Agarwal, R., Schwarzer, M., Castro, P.S., Courville, A. and Bellemare, M.G., 2021. Deep reinforcement learning at the edge of the statistical precipice. In NeurIPS. https://openreview.net/forum?id=uqv8-U4lKBe

---

> ### Author Response · Authors · 2021-11-13
> **Response for reliable evaluation**
>
> Hi, Rishabh
>
> Thanks for your suggestions on reliable evaluation. We follow your NeurIPS paper and the released code to evaluate the performance in D4RL benchmark. We compare the Bootstrap CIs, Performance Profiles, and Aggregate Metrics (including interquartile mean and optimality gap) for all methods. The results are given in [1]. We will add the results as an appendix in the final version.
>
> [1] https://www.dropbox.com/s/ufhu5gdvwf86la0/Reliable-Evaluation-PBRL.pdf?dl=0

---

> > ### Comment · Area_Chair_oN9s · 2021-11-20
> > **Re: reliable evaluation**
> >
> > Dear authors,
> >
> > Thank you for the additional results.  You should be able to update the pdf on this site.
> >
> > I also would appreciate it if you could respond to questions and concerns that reviewers have given.

---

> > > ### Author Response · Authors · 2021-11-23
> > > **Response to Area Chair oN9s**
> > >
> > > Dear Area Chair,
> > >
> > > Thanks for your response. We have given detailed responses to the reviewers’ questions. We also included a summary of the paper revision in the beginning.

---

### Author Response · Authors · 2021-11-23
**Summary of paper updates**

Paper updates:

- **[Section 3]** We revised Lemma 1 to an informal Claim, clarified the assumptions of theoretical analysis, and added discussions of OOD sampling procedures.

- **[Section 4]** We discussed related works of Bellman completeness assumption, KNR, BOPAH and EDAC.

- **[Appendix B]** We clarified the hyper-parameter settings, added the computation comparison, and gave a remark on formulations of pessimism in actor and critic.

- **[Appendix C]** We clarified the evaluation method, and added ablation studies on OOD sampling and actor training.

- **[Appendix E]** We added reliable evaluation to address the statistical uncertainty in experiments.

We greatly appreciate all reviewers' suggestions. We hope that our revisions address the reviewers' concerns. Please let us know if there are further questions.

---

> ### Comment · Reviewer_hc93 · 2021-11-25
> **What does PEVI on Figure 18, 19 mean?**
>
> In Figures 18 and 19 from the updated version, I think there are some typos. (what does PEVI mean for?)

---

> > ### Author Response · Authors · 2021-11-25
> > **Thanks for pointing out this typo**
> >
> > We thank the reviewer for pointing out this problem. The name "PEVI" and "PEVI-Prior" in Fig. 18 and Fig. 19 should be "PBRL" and "PBRL-Prior", respectively. PEVI is the abbreviation of the Pessimistic Value Iteration algorithm [1], which is a stylized algorithm for linear MDPs (see Appendix A). We will correct this typo in the final version.
> >
> > [1] Ying Jin,  Zhuoran Yang, and Zhaoran Wang. Is Pessimism Provably Efficient for Offline RL?. In ICML 2021

---

### Decision · Program_Chairs · 2022-01-20

**Decision:**

Accept (Spotlight)

**Comment:**

This paper makes significant advances in offline reinforcement learning by proposing a new approach of being pessimistic to deal with uncertainties in the offline data.  The proposed approach uses bootstrapped Q-functions to quantify the uncertainty, which by itself is not new, and introduces additional data based on the pseudo-target that is penalized by the uncertainty quantification.  The use of such additional data is the first of a kind, and the paper provides theoretical support for the case of linear MDP and empirical support with the D4RL benchmark.  The reviewers had originally raised concerns or confusions regarding theoretical analysis and experiments.  The authors have well responded to them, and no major concerns remain.